# Stress-induced nuclear translocation of ONAC023 improves drought and heat tolerance through multiple processes in rice

Yu Chang[1,2], Yujie Fang [3]✉, Jiahan Liu[1,2], Tiantian Ye[1,2], Xiaokai Li[1,2], Haifu Tu[1,2], Ying Ye[1,2], Yao Wang[1,2] & Lizhong Xiong [1,2]✉

Drought and heat are major abiotic stresses frequently coinciding to threaten rice production. Despite hundreds of stress-related genes being identified, only a few have been confirmed to confer resistance to multiple stresses in crops. Here we report ONAC023, a hub stress regulator that integrates the regulations of both drought and heat tolerance in rice. *ONAC023* positively regulates drought and heat tolerance at both seedling and reproductive stages. Notably, the functioning of *ONAC023* is obliterated without stress treatment and can be triggered by drought and heat stresses at two layers. The expression of *ONAC023* is induced in response to stress stimuli. We show that overexpressed ONAC23 is translocated to the nucleus under stress and evidence from protoplasts suggests that the dephosphorylation of the remorin protein OSREM1.5 can promote this translocation. Under drought or heat stress, the nuclear ONAC023 can target and promote the expression of diverse genes, such as *OsPIP2;7*, *PGL3*, *OsFKBP20-1b*, and *OsSF3B1*, which are involved in various processes including water transport, reactive oxygen species homeostasis, and alternative splicing. These results manifest that ONAC023 is fine-tuned to positively regulate drought and heat tolerance through the integration of multiple stress-responsive processes. Our findings provide not only an underlying connection between drought and heat responses, but also a promising candidate for engineering multi-stress-resilient rice.

Human population is on track to reach 10 billion by 2060, demanding that the global rice yields correspondingly increase by 1.0%–1.2% per year to narrow the food supply deficit[1,2]. Despite numerous efforts taken to improve rice yield in the past decades, unfavorable climate conditions, led by drought and heat, remain serious threats to rice production. It was estimated that drought can result in more than 45% yield loss of rice at the reproductive stage[3], and heat can cause a 10% reduction in rice grain yield for every 1 °C rise in night temperature[4]. Especially for *japonica* rice, air temperatures exceeding 35 °C in field often cause a significant yield penalty[5,6]. Increasing concern is that the

continuous global warming driven by greenhouse gas emissions will increase the intensity, frequency, and duration of the stress combination of drought and heat in the years ahead [http://www.ipcc.ch/]. Therefore, deciphering the mechanisms of plant response and adaptation to drought and heat stress is expected to provide opportunities to breed multi-stress-resilient crops to guarantee global food security.

NAC (NAM, ATAF and CUC) is a well-studied TF family with many members participating in plant stress responses[7]. For example, *STRESS-RESPONSIVE NAC 1* (*SNAC1*), predominantly expressed in guard cells under drought stress, confers drought resistance at vegetative

[1]National Key Laboratory of Crop Genetic Improvement, Huazhong Agricultural University, Wuhan 430070, China. [2]Hubei Hongshan Laboratory, Huazhong Agricultural University, Wuhan 430070, China. [3]Key Laboratory of Plant Functional Genomics of the Ministry of Education, Yangzhou University, Yangzhou 225009, China. ✉e-mail: yjfang@yzu.edu.cn; lizhongx@mail.hzau.edu.cn

and reproductive stages by promoting stomatal closure in rice[8]. Later, group II and III NACs such as SNAC2/OsNAC6, ONAC045, SNAC3, OsNAC2, ONAC127, and ONAC129 have been identified as contributors to drought, salt, cold, and/or heat stress tolerance[9–15]. However, few of the reported stress-resilient NACs confer both drought and heat tolerance in rice. Especially, the effects on stress resistance improvement of most reported multi-stress-resilient NACs have not been systematically evaluated in field conditions. Despite the great efforts made in stress-resilient NAC characterization, these concerns restrict the application of NAC genes in stress-resilient rice breeding.

As transcription factors, NAC proteins generally function in the nucleus. A special category of NAC members can anchor to the plasma or endoplasmic reticulum membrane and translocate to the nucleus by stimulation-dependent cleavage of the transmembrane domain. This membrane-nucleus translocation is so far the only known mechanism for NAC relocation, which was revealed by studying the subcellular location of stress-responsive NACs, such as ANAC013, ANAC062, OsNTL3, and ONAC054[16–19]. Thus, the functions of these NACs are more stringently controlled, and unlikely to cause leaking effects when the stimulation signal is absent. However, factors governing the cleavage and relocation processes remain largely unknown.

In this study, we identified ONAC023 as a putative hub regulator for drought and heat stress responses, conferring drought and heat tolerance at both vegetative and reproductive stages in rice. We found that drought and heat stresses not only induced *ONAC023* transcript abundance but also triggered the nuclear translocation of ONAC023 with the assistance of a remorin protein OsREM1.5. ONAC023 regulates drought and heat tolerance by targeting diverse categories of genes involved in multiple stress-responsive processes such as redox homeostasis, water transport, and alternative splicing regulations. These findings reveal a multi-layered regulation integrated by NAC in response to drought and heat stresses, and present *ONAC023* as a valuable source gene for breeding drought- and heat-resilient rice.

## Results

### Identification of a hub stress-responsive NAC gene associated with drought and heat tolerance in rice

To efficiently identify NAC genes responsive to drought and heat stress, we developed a rapid evaluation workflow that integrates the transcriptome, phenome, and genome data (Fig. 1a)[20,21]. A NAC-centered gene regulatory network (GRN) were established using the differentially expressed genes (DEGs) under drought and heat stresses compared to the normal conditions from the transcriptome sequencing (RNA-seq) data of two *japonica* varieties, Zhonghua 11 (ZH11) and Dongjin (DJ), at seedling and reproductive stages (Fig. 1b and Supplementary Fig. 1a, b). The DEGs were grouped into nine clusters (C1–C9, Supplementary Fig. 1c), using an independent component analysis (ICA) algorithm[22], and a regression-tree-based network was applied to infer the regulatory partnership between the NACs and DEGs within each cluster. Hub NACs with the highest degree of connectivity were identified as potential drought- and heat-responsive NACs (Fig. 1b).

Among nine hub regulators, ONAC129 and ONAC045 have been reported to confer heat and drought tolerance, respectively[9,11], indicating the effectiveness of the GRN. To further evaluate the importance of hub NACs in stress resistance, the genomic variations within the gene body and promoter regions of the nine NACs in 210 rice germplasms were extracted from RiceVarMap2.0[21] and subjected to genetic association analysis with our previously reported image-based traits (i-traits) as indicators of drought tolerance (Supplementary Data 1)[20]. *ONAC023*, from subgroup V of the NAC family[23], was identified with significant associations of three single nucleotide polymorphisms and one insertion/deletion (InDel) with "Relative Projected Green Area Ratio" (GPAR-R, data applicable for 180 varieties), a critical i-trait well reflecting drought tolerance[20] (Fig. 1c, Supplementary

Fig. 2a and Supplementary Data 2). The effect of a specific genetic variation (InDel *vg6402578*) in the promoter region of *ONACO23* on the transcriptional activity of the promoter was examined by dual-luciferase (dual-LUC) assay. The result showed that the transcriptional activity of the promoter with the presence of the InDel (InDel$^+$) was ~2.5 folds higher than that of the promoter without the InDel (InDel$^-$) (Fig. 1c). Considering the *cis*-regulatory effect of the InDel and its loose linkage disequilibrium (LD), we arbitrarily divided the rice populations into InDel$^-$ and InDel$^+$ groups to evaluate the genetic effect of the InDel. The InDel$^+$ group showed significantly higher GPAR-R than the InDel$^-$ group ($P = 1.2e - 09$ by two-tailed Student's $t$ test, Fig. 1d). Quantitative reverse transcription PCR (RT−qPCR) analysis on 98 varieties under drought stress revealed that the InDel$^+$ group exhibited significantly higher *ONACO23* expression levels than the InDel$^-$ group ($P = 0.025$ by two-tailed Student's $t$ test, Fig. 1e). In addition, *ONACO23* expression was significantly induced under drought and heat stress (~4 folds within 5.5-h treatment, Fig. 1f) in ZH11 (InDel$^+$ genotype). Besides, analyses on the variations' geographic distribution patterns, variety−genotype composition, and nucleotide diversity suggested a selective sweep around the *ONACO23* locus, particularly in *japonica* rice during the selection for climate adaptation (Supplementary Fig. 2b−e). Overall, these findings demonstrate that *ONACO23* may be a critical regulator of drought and heat stress responses in rice.

### ONAC023 confers drought tolerance at vegetative and reproductive stages

To validate whether ONAC023 functions in regulating drought resistance, a T-DNA insertion mutant (*onac023*) was obtained, and mutant-complementation plants (*onac023 ONAC023*) were generated. The survival rates after recovery from the drought stress indicated that *onac023* was more sensitive to drought stress compared to the DJ-WT and *onac023 ONAC023* plants (Fig. 2a, Supplementary Fig. 3a). We further evaluated their agronomic trait performance along with two RNA interference lines and ZH11-WT in the field under drought conditions. After severe drought stress (relative soil moisture (RSM) < 10% at the booting stage) followed by recovery (RSM = 100%), *onac023* and *ONAC023*-RNAi lines exhibited drought-sensitive phenotypes including obvious tissue death, impaired growth, and over 2 folds reductions in multiple agronomic traits including yield and biomass compared with the *onac023 ONAC023* or the corresponding WT lines (Fig. 2b, Supplementary Fig. 4). In contrast to the WT lines, the *onac023* and *ONAC023*-RNAi lines showed significantly higher relative electrolyte leakage and free proline contents but significantly lower relative water content, indicating that the leaf tissue of the *onac023* and *ONAC023*-RNAi lines suffered more severe damages during the drought stress treatments (Supplementary Fig. 5). These results suggested that *ONAC023* is indispensable for drought tolerance in rice.

To investigate whether *ONAC023* is efficient for drought tolerance improvement, transgenic lines constitutively overexpressing *ONAC023* without or with a C-terminal FLAG tag, termed *ONAC023*-overexpressing (*ONAC023*-OE) and *ONAC023−FLAG*-overexpressing (*ONAC023−FLAG*-OE) constructs, respectively, were generated. The i-traits of two *ONAC023*-OE lines and the WT (ZH11) plants were tested for drought tolerance at booting stage. The *ONAC023*-OE lines exhibited significantly higher GPAR-R than ZH11 during the drought treatment ($P < 0.05$ by ANOVA−HSD test, Fig. 2c), consistent with the positive correlation between *ONAC023* expression and GPAR-R values (Fig. 1c−e). A multi-location drought tolerance test conducted in Wuhan (Hubei province, PRC) and Lingshui (Hainan province, PRC) revealed that *ONAC023* overexpressors consistently exhibited increased drought tolerance in terms of biomass and yield compared to ZH11 (Fig. 2d, e, and Supplementary Fig. 6). Moreover, overexpression and mutation of *ONAC023* had minor effects on the major agronomic traits under normal growth conditions ($P > 0.05$ by ANOVA−HSD test, Fig. 2f and Supplementary Fig. 7). These results

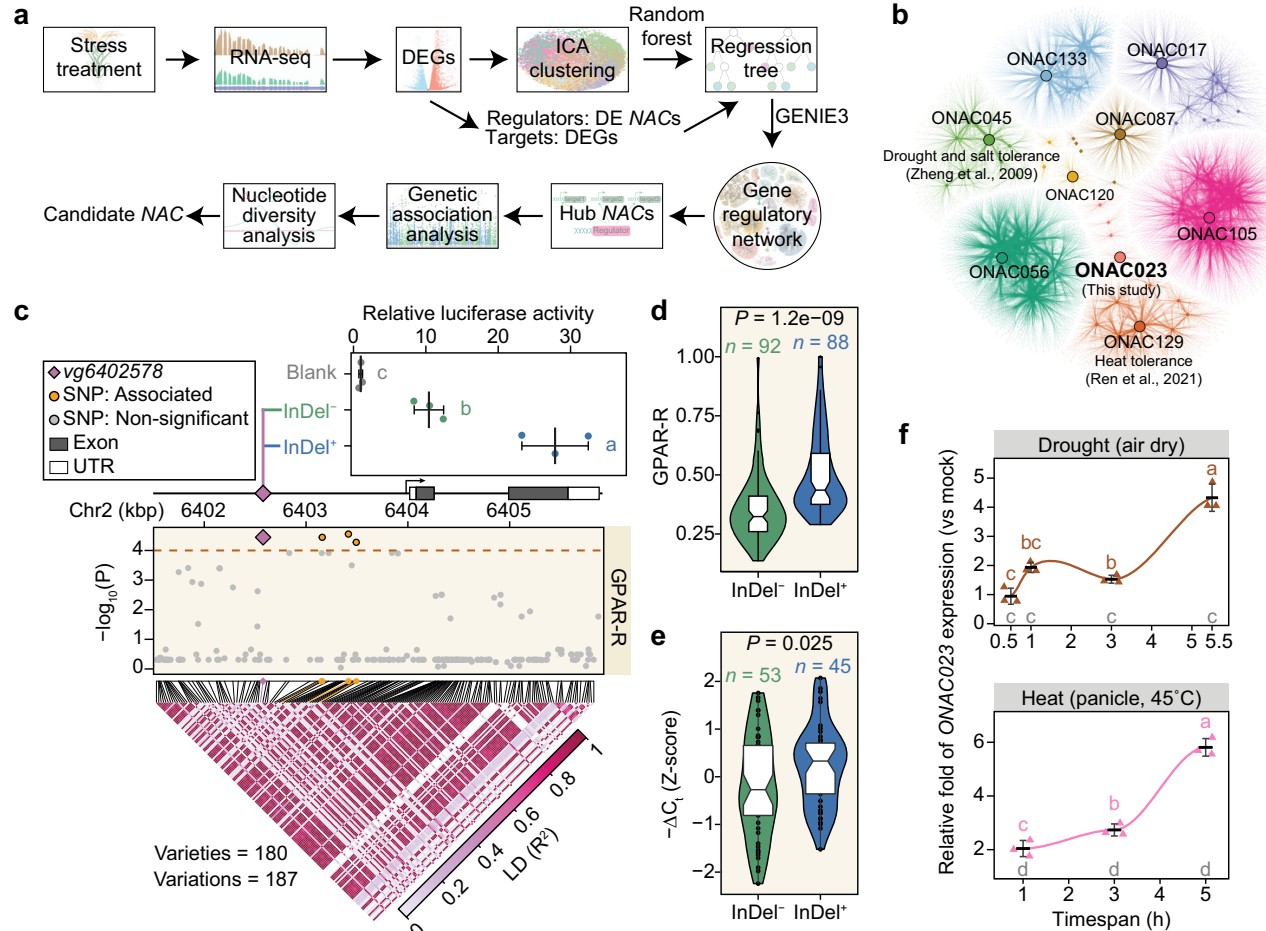

**Fig. 1 | Integrative analysis of rice NAC genes with potential functions in stress tolerance. a** Overview of the selection work flow applied to rice NAC genes. **b** NAC-centered regulator network in response to drought and heat stresses. Each edge represents a proposed NAC–target relation. ICA clusters are denoted by different colors. The hub NACs are labeled. **c** Association analysis of *ONAC023* with GPAR-R. The Manhattan plot illustrates the power of associations between genetic variations and GPAR-R (in terms of the *F* test *P* values). The triangle heatmap shows the LD. The *ONAC023* promoter activities with/without the insertion of *vg2402578* (termed as InDel⁺ and InDel⁻) evaluated by dual-LUC assay were expressed as the relative fluorescence intensity of fLUC/rLUC, and all values were normalized against that of the blank construct. **d** Significant difference in the GPAR-R between the InDel⁺ and InDel⁻ variety groups. **e** *ONAC023* expression among the 98 tested rice varieties under drought stress. The mean RT–qPCR −$\Delta C_t$ values (determined by three plants for each variety) were normalized as Z-scores. The lower, central, and upper hinges

of the box plots in (**d, e**) correspond to the 25th, 50th (median), and 75th percentiles, respectively. The whiskers extend from the hinges to the largest/smallest values no further than 1.5× IQR (inter-quartile range) from the hinges. **f** *ONAC023* expression in ZH11 under drought and heat stress treatments. The expression of *ONAC023* was normalized to that of the corresponding mock samples. The curves indicate the smoothed conditional means by LOESS regression. The significance levels of the treated samples were labeled at the upper side, while those of the mocks were labeled (in gray) at the bottom of each panel. The cross-bars in (**c, f**) represent mean ± SD (standard deviation). Letter symbols denote significances ($P < 0.05$) determined by one-way ANOVA (two-sided) with Tukey's Honest significant difference (HSD) test. Exact *P* values and details of the one-way ANOVA–HSD test can be found in the Source Data file. *P* values in (**d, e**) were determined by two-tailed Student's *t* test. Source data are provided as a Source Data file.

together indicate that *ONAC023* confers drought tolerance at both vegetative and reproductive stages.

## ONAC023 confers heat tolerance at vegetative and reproductive stages

Since *ONAC023* expression was also induced by heat stress (Fig. 1f), we investigate its potential function in heat tolerance. We found that after a 45 °C heat stress treatment at the seedling stage, *onac023* had lower survival rates compared with the DJ-WT and *onac023 ONAC023*, while the *ONAC023*-OE lines showed over 5 folds higher survival rates compared to ZH11-WT (Fig. 3a, Supplementary Fig. 3b). At the reproductive stage, *onac023* had reduced seed-setting rate compared to DJ-WT after a 43 °C heat stress treatment, while *ONAC023*-OE lines showed less damage and a tenfold higher seed-setting rate compared to ZH11-WT (Fig. 3b). Pollen germination capability on the post-flowering stigmas,

an indicator for the vulnerability of rice fertility to high temperatures[24], was also assessed under heat stress (43 °C) and normal conditions (30 °C). *onac023* showed a significantly lower relative pollen germination ratio (heat vs normal) compared to DJ-WT, while *ONAC023*-OE lines showed a higher ratio than ZH11-WT (Fig. 3c). These results indicate that ONAC023 also confers heat tolerance in terms of pollen fertility.

In addition, results from a two-year field test in Wuhan, where natural heat stress occurred with a peak air temperature of 35–45 °C during the heading stage (Supplementary Fig. 8), showed that *ONAC023*-OE lines exhibited less yield penalty and seed-setting rate reduction than ZH11-WT (Fig. 3d, e). The *ONAC023–FLAG*-OE lines also had ~2 folds higher seed-setting rate and yield (Supplementary Fig. 9). These results suggest that *ONAC023* confers heat tolerance and has high potential for heat tolerance improvement.

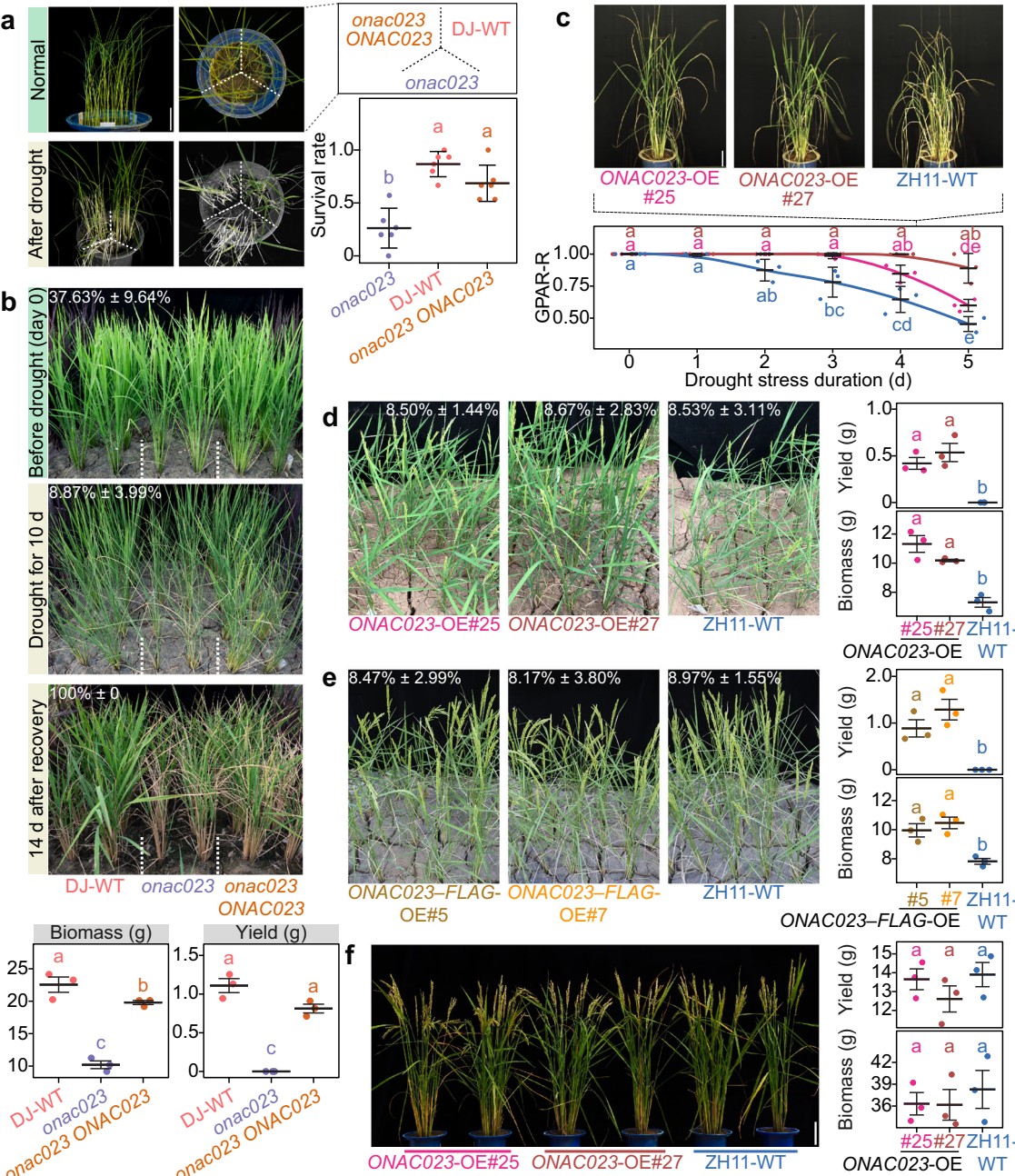

**Fig. 2 | *ONAC023* confers drought tolerance in rice. a** Phenotypes of the T-DNA insertion mutant *onac023*, the derived negative control (DJ-WT), and the complementation (*onac023 ONAC023*) seedlings after 7-d recovery from the drought stress treatment (RSM < 10% for 7 d). One representative photo (out of six buckets under independent treatments) is shown. Green tissues after recovery were highlighted. See Supplementary Fig. 3a for the raw photos. Scale bar = 5 cm. Cross-bars represent mean ± SEM (standard error of the means). **b** Phenotypes of one of the three blocks of *onac023*, DJ-WT, and *onac023 ONAC023* during the in-field drought stress at the booting stage. The top-left white number in each photo indicates RSM (mean ± SD, *n* = 3). The biomass and yield (per plant) data is presented as mean ± SEM in the bottom cross-bar plots. **c** Phenotypes of the *ONAC023*-OE lines under drought stress. Plants were monitored every day by the phenotyping platform starting from the day 4 after drought stress initiation (day 0 in the *X*-axis). Scale

bar = 10 cm. Cross-bars denote mean ± SD of three plants with independent treatments. The curves indicate the smoothed conditional means by LOESS regression. **d**, **e** In-field drought tolerance evaluation of the *ONAC023* overexpressors. Phenotype of one repeat under drought stress is shown for each transgenic line. RSM is labeled as mean ± SD. The plots on the right show the statistics on the biomass and yield (per plant) of the tested lines. Cross-bars denote mean ± SEM of the three randomized blocks. **f** Agronomic traits of the *ONAC023* overexpressors grown under normal growth conditions at the ripening stage. Scale bar = 10 cm. The cross-bar plot shows the mean ± SEM of yield (per plant) and biomass based on three randomized blocks (at least six plants each). Significances (*P* < 0.05) in this figure were determined by one-way ANOVA–HSD test (two sided). Exact *P* values and details of the one-way ANOVA–HSD test can be found in the Source Data file. Source data are provided as a Source Data file.

## Nuclear localization of ONAC023 is induced by drought and heat stresses

To unveil the regulatory mechanisms of ONAC023 in improving drought and heat tolerance, we first investigated its subcellular location in *ONAC023–FLAG*-OE#5 leaf tissues by immunofluorescence

assay. The result showed that ONAC023–FLAG was largely localized outside the nuclei under the normal conditions (25 °C with ample watering, Fig. 4a and Supplementary Fig. 10). However, after drought (RSM < 10 % for 3 d) or heat (45 °C for 3 h) stress, ONAC023–FLAG was significantly enriched in the nuclei (Fig. 4a and Supplementary Fig. 10).

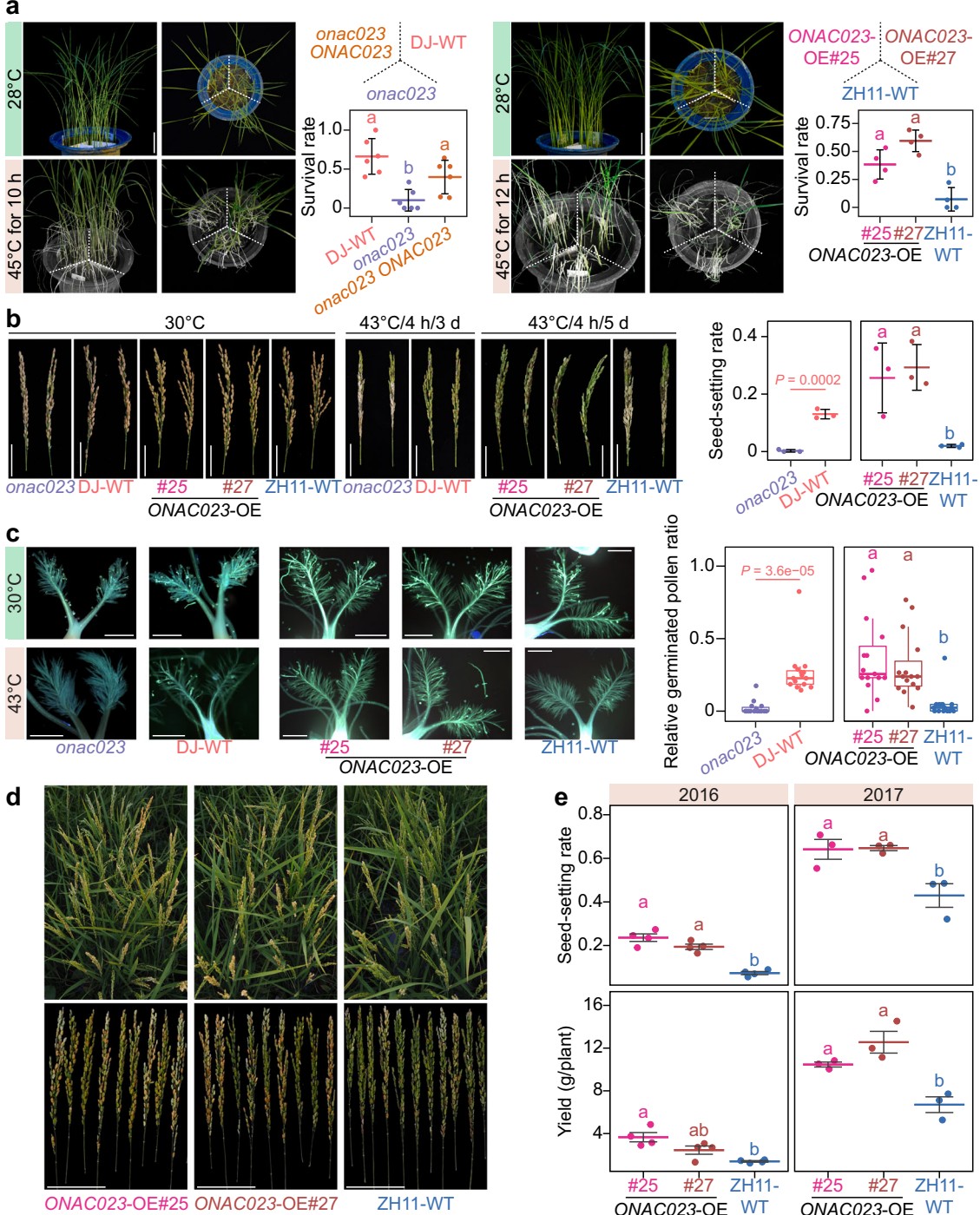

**Fig. 3 | *ONAC023* confers heat tolerance in rice. a** Heat tolerance evaluation of *onac023* mutant and *ONAC023* overexpressors. Green tissues after recovery were highlighted. See Supplementary Fig. 3b for the raw photos. Scale bars = 5 cm. Cross-bars represent mean ± SD of the survival rates (six and four independently treated buckets for *onac023* and *ONAC023* overexpressors, respectively). **b** Heat tolerance evaluation of *onac023* mutants and *ONAC023* overexpressors in terms of spikelet fertility. Representative panicles from two plants are shown for each line. Scale bars = 5 cm. Seed-setting rates of the heat stress-treated plants after recovery are shown by the cross-bars (mean ± SD, three plants with independent treatments). **c** Aniline blue staining of the germinated pollens on the heat stress-treated and untreated (mock) stigmas. Scale bars = 0.5 mm. The relative germinated pollen ratios (heat vs mock) of the tested lines were illustrated by boxplot (15 plants for

each line). Whiskers of the box plot illustrate the 1.5× IQR ranges, and hinges denote the 25th and 75th percentiles. **d** Plant and panicle phenotypes of one representative block of the *ONAC023*-OE plants at the ripening stage, suffered high temperature (peak air temperature exceeded 40 °C during daytime) in the field. Scale bars = 10 cm. **e** Heat tolerance improvement of *ONAC023*-OE lines under the field conditions with heat stress occurred in August at Wuhan. Dots indicate the average values of six plants in each block. Cross-bars denote mean ± SEM. The air temperatures during heat stress are shown in Supplementary Fig. 8. *P* values between *onac023* and DJ-WT were determined by two-tailed Student's *t* test in (**b**, **c**). Multi-group comparisons in (**a–c**, **e**) were evaluated by one-way ANOVA–HSD test (two-sided) at *P* < 0.05. Exact *P* values and details of the one-way ANOVA–HSD test can be found in the Source Data file. Source data are provided as a Source Data file.

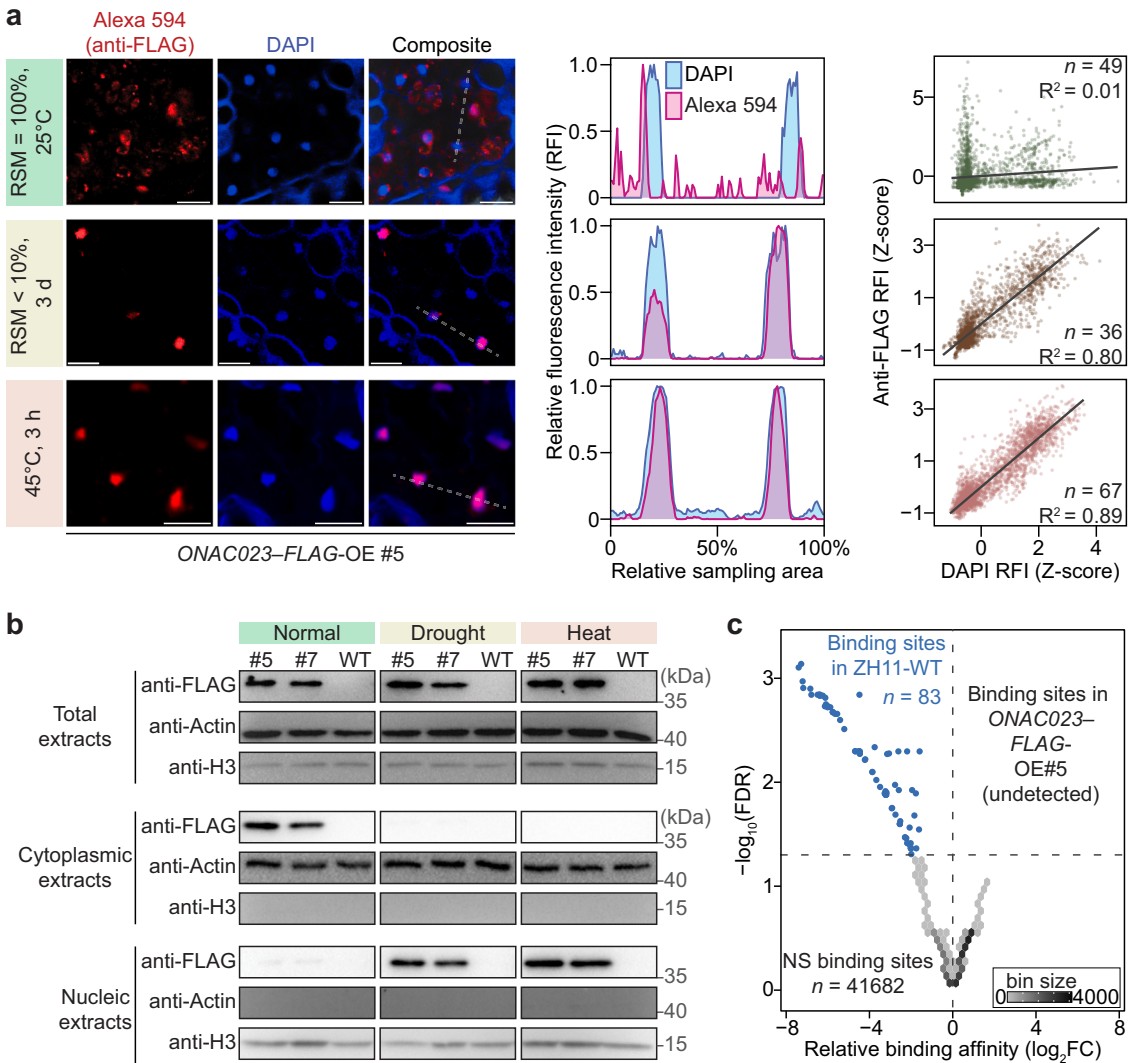

**Fig. 4 | Drought and heat stress-induced nuclear localization of ONAC023. a** The microscopic immunofluorescence images show the subcellular localization of ONAC023−FLAG in the seedling tissues. The anti-FLAG signal (by Alexa 594) and nuclei stained by DAPI were shown in red and blue, respectively. Scale bars = 10 μm. The Alexa 594 and DAPI signal distribution profiles are illustrated by the peak plots in the middle. Relative fluorescence intensity (RFI), representing the percentage fluorescence intensity to the maximum intensity in the sampling region, was quantified to evaluate the fluorescence distribution. The sampled regions were marked with dashed lines. The scatter plots (on the right side) summarize the immunofluorescence co-localization of all assayed slices. Each point represents a measured pixel in terms of the Z-transformed DAPI and anti-FLAG RFIs measured in each cell. The slopes show the fitted linear models by the two RFIs. Overall co-localization under each condition was assessed by the $R^2$ value of Pearson correlation coefficient (PCC) analysis. $n$ denotes the number of observed cells in the quantification. **b** Subcellular localization of ONAC023 under drought stress, heat stress, and normal conditions was confirmed by Western blot with an anti-FLAG antibody. Actin and histone H3 were set as endogenous controls for cytoplasmic proteins and nuclear proteins, respectively. See Supplementary Fig. 23 for the uncropped images. **c** ChIP−seq analysis of *ONAC023−FLAG*-OE#5 and ZH11-WT seedlings under normal conditions. Sequenced peak regions with no significant difference (NS, by the threshold of $\log_2FC > 1$ and FDR < 0.05) between the two samples are indicated as hex-bins (gray), while regions enriched in the ZH11-WT sample (possible background) are shown as blue points. The FDR values were determined by Wald test (two-sided) with Benjamini-Hochberg correction. Source data are provided as a Source Data file.

The nuclear and cytoplasmic proteins from the *ONAC023−FLAG*-OE leaf tissues under the same treatments were further examined by Western blot. The result showed that the cytoplasmic ONAC023−FLAG was more abundant under the normal conditions compared to drought and heat treatments, while nuclear ONAC023−FLAG exhibited the opposite trend (Fig. 4b). A chromatin immunoprecipitation−sequencing (ChIP−seq) assay using the seedling leaf tissue from *ONAC023−FLAG*-OE#5 and ZH11-WT was performed to investigate the DNA-binding function of ONAC023−FLAG in the nuclei under normal conditions. Even though the ONAC023−FLAG protein was constitutively expressed in the *ONAC023−FLAG*-OE#5 under the normal condition (25 °C, Supplementary Fig. 11a), no binding of

ONAC023−FLAG in *ONAC023−FLAG*-OE#5 was detected (with a threshold of 2-fold change and FDR < 0.05 by Wald test, Fig. 4c). These results together suggest that the stress-induced nuclear translocation of ONAC023 may be a strategy for its functional activation under stress conditions, in addition to stress-induced expression at the transcriptional level (Fig. 1f).

## OsREM1.5 is required for the stress-induced nuclear translocation of ONAC023

Since ONAC023 lacks the transmembrane domain and intact nuclear-localization signal sequence that are required for nuclear translocation of reported membrane NACs (Supplementary Fig. 12), suggesting a

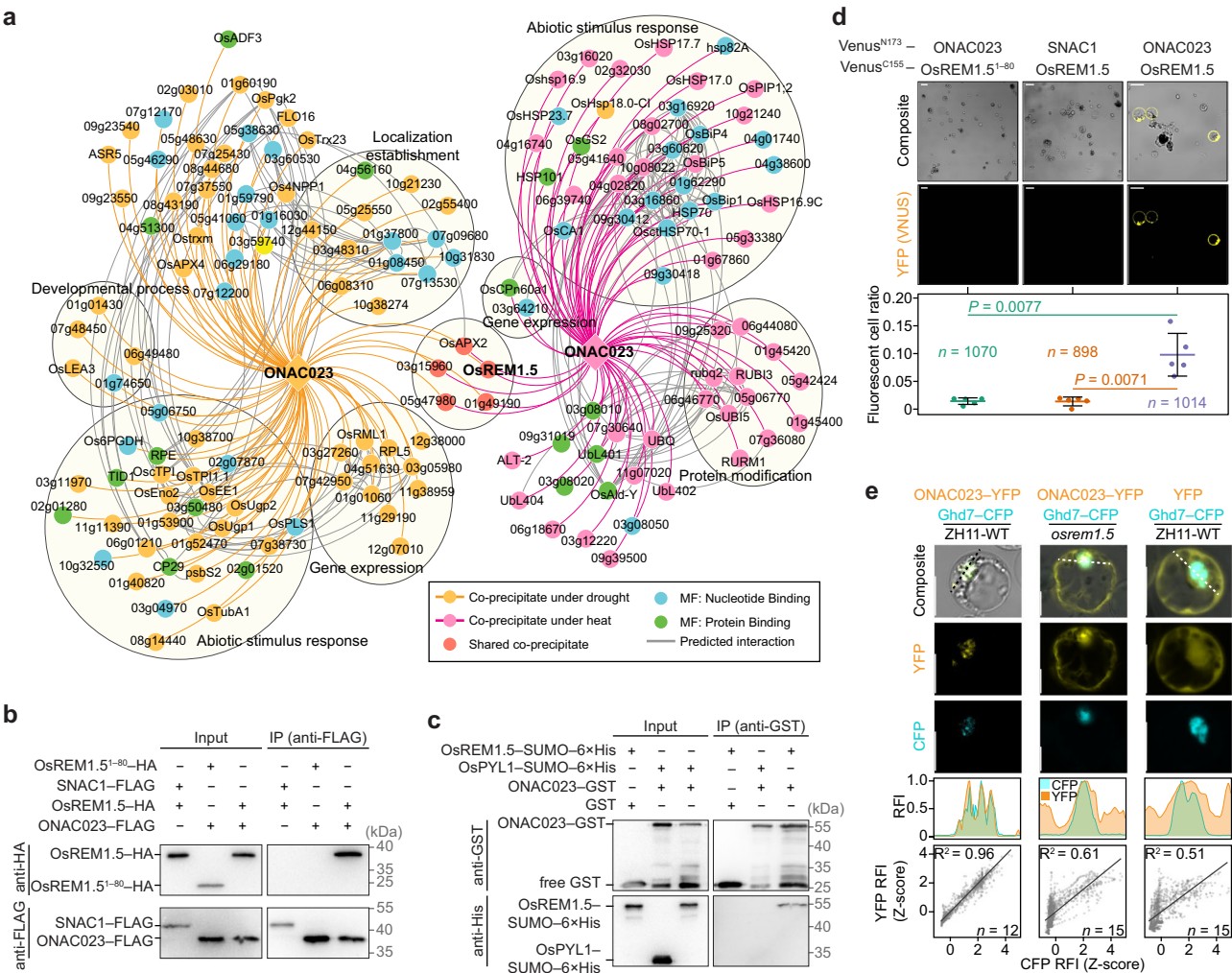

**Fig. 5 | Identification of the co-factor for the nuclear transport of ONAC023 under the stress conditions. a** Integrated network of the ONAC023-interacting candidates identified by IP-MS using drought and heat stress-treated *ONAC023–FLAG*-OE#5 transgenic plants. Gray curves link the predicted high-confidence interactions by STRING program. Reported proteins were labeled by their names, while the rest were marked by accession IDs. MF, molecular function. Validation of the ONAC023-OsREM1.5 interaction by Co-IP (**b**), in vitro pull-down assay (**c**), and BiFC (**d**). In Co-IP and BiFC assays, an irrelevant NAC protein (SNAC1) and the 1–80 amino acid (AA) truncated form of OsREM1.5 (OsREM1.5^1–80) were used as negative controls. In the pull-down assay, an irrelevant protein (OsPYL1) and free GST were used as negative controls. The cross-bar plot in (**d**) shows the ratio of fluorescent cells (mean ± SD) in five repeats of the assay. *P* values were evaluated by two-tail Student's *t* test. Scale bars = 20 μm. See Supplementary Fig. 23 for the

uncropped images for (**b**, **c**). **e** Slices showing the subcellular localization of ONAC023 (tagged by YFP) in the WT (variety ZH11) and *osrem1.5* protoplasts. Empty YFP was expressed as a control, and Ghd7–CFP was co-transfected as a nuclear marker. Scale bars = 10 μm. The middle peak plots illustrate the profiles of the ONAC023–YFP and Ghd7–CFP relative fluorescence intensities. Sampled regions are marked with dashed lines in the slices. The bottom scatter plots show the result of co-localization analysis of all assayed cells in each transfection. Each point represents a measured pixel in terms of YFP and CFP RFIs (normalized by Z-transform for each cell). The fitted linear models by two-color fluorescence are expressed as the slopes. The R² value of PCC was used to illustrate the overall YFP–CFP co-localization in all observed cells. Source data are provided as a Source Data file.

unique mechanism for ONAC023 which is different from the current transmembrane domain-cleavage model[18]. To reveal the mechanism of the stress-induced nuclear enrichment of ONAC023, we performed anti-FLAG immunoprecipitation coupled with tandem mass spectrometry identification (IP-MS) using protein extracts from drought (RSM < 10% for 3 d) and heat (45 °C for 8 h) stress-treated *ONAC023–FLAG*-OE#5 and ZH11-WT seedlings. Although only five immunoprecipitated proteins were detected in both conditions (91 for drought stress and 69 for heat stress, Supplementary Data 3), the proteins with the GO term "Abiotic stimulus response" were significantly enriched in both drought-treated (31 proteins) and heat-treated (37 proteins) samples (Fig. 5a), implicating the involvement of potential ONAC023-interacting proteins in drought and/or heat stress responses. We further focused on the five overlapped immunoprecipitated proteins to identify candidates responsible for ONAC023

nuclear translocation. The interaction between ONAC023 and remorin protein OsREM1.5 was subsequently confirmed by co-immunoprecipitation (Co-IP), bimolecular fluorescence complementation (BiFC) in ZH11 protoplasts, and in vitro pull-down assays (Fig. 5b–d). Remorin protein was reported to be involved in nuclear translocation process in *Arabidopsis thaliana* by interacting with importin-α proteins[25]. To further validate if OsREM1.5 is required for ONAC023 nuclear translocation, we examined the ONAC023 localization in ZH11 protoplasts with *OsREM1.5* mutated by CRISPR-Cas9 technique (Supplementary Fig. 13). Although ONAC023 was also detected in the nuclei of native ZH11 protoplasts, which may be due to uncertain stress factors during the transfection and/or the unspecific nuclear localization of the YFP tag (as is revealed by the control YFP transfection, Fig. 5e), the mutation of *OsREM1.5* resulted in a significantly higher accumulation of ONAC023–YFP in cytoplasm

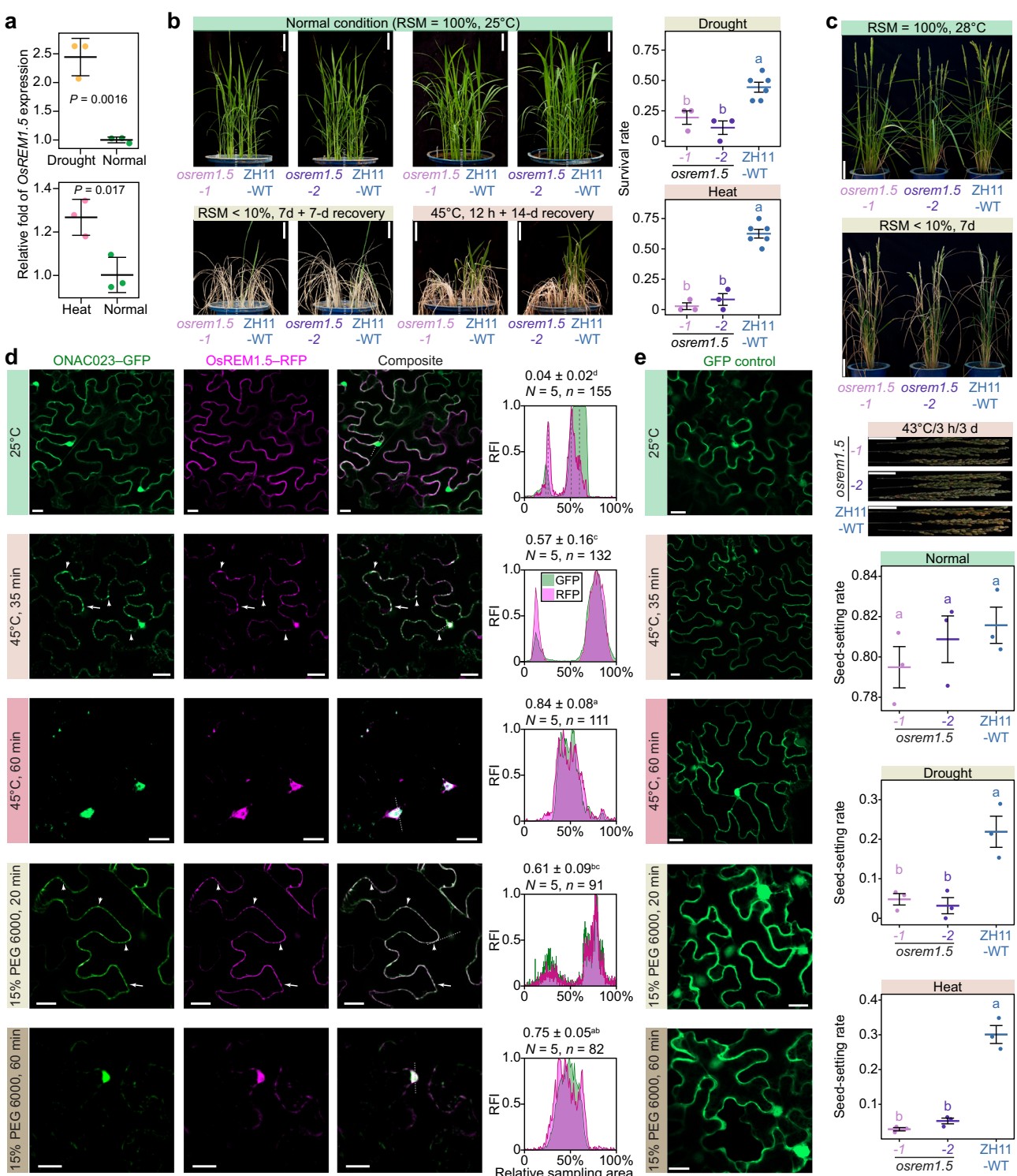

compared to that in the wild-type (WT) control (Fig. 5e). These results demonstrated that OsREM1.5 is required for an efficient nuclear translocation of ONAC023. In addition, OsREM1.5 was verified to interact with two importin-α proteins (OsIMP-α1a and OsIMP-α1b) by BiFC (Supplementary Fig. 14a), which was similar to the results in *Arabidopsis*[25]. These results suggest that OsREM1.5 functions in stress-induced nuclear accumulation of ONAC023 likely through the OsREM1.5–improtin-α-mediated nuclear translocation.

Furthermore, *OsREM1.5* was found to be transcriptionally induced by drought and heat stresses (Fig. 6a), and its mutation led to decreased drought and heat tolerance, as indicated by significantly

lower seedling survival rates and seed-setting rates of two CRISPR-Cas9-mediated mutant lines compared to ZH11-WT (*P* < 0.05 by ANOVA–HSD test, Fig. 6b, c). To examine whether OsREM1.5-assisted nuclear translocation of ONAC023 is an abiotic stress-dependent process, drought (by 15% PEG 6000) and heat (45 °C) stress treatments on *Nicotiana benthamiana* (*N. benthamiana*) were applied for tracing the subcellular localization of ONAC023 and OsREM1.5 in the epidermal cells. Under the normal conditions, OsREM1.5–RFP and ONAC023–GFP were observed to be evenly distributed on the plasma membrane and in the cytosol, respectively (Fig. 6d). Though ONAC023–GFP was observed in the nucleus under normal conditions,

**Fig. 6 | OsREM1.5 is responsive to drought and heat stresses. a** The relative transcription fold change of *OsREM1.5* under the drought or heat stress treatment compared to the non-stress condition. Cross-bars represent mean ± SD of three repeats of the treatments (at least 10 seedlings each). *P* values were determined by two-tailed Student's *t* test. Drought and heat tolerance evaluations on the *osrem1.5* mutants at the seedlings stage (**b**) and flowering stage (**c**). The survival rates of the recovered seedlings and the seed-setting rates of the recovered panicles (20 d after the treatment) were exhibited by cross-bar plots (mean ± SD). Three (for each mutant line) or six (for the ZH11-WT) buckets of seedlings and three flowering plants (for each mutant line and ZH11-WT) were tested with independent treatments. Scale bars = 5 cm (**b**) and 10 cm (**c**). **d** Dynamic subcellular localization changes of ONAC023–GFP and OsREM1.5–RFP in response to drought or heat stress treatment. *N. benthamiana* epidermal cells under 45 °C for 0 min, 35 min, and 60 min or treated with 15% PEG 6000 (drought stress) for 20 min and 60 min were

imaged. Typical plaque-like localizations are labeled by arrows. The peak plots illustrate the profiles of the GFP and RFP relative intensities. Differential localization of ONAC023–GFP (cytoplasm) and OsREM1.5–RFP (cell membrane) was indicated by the dashed lines that show the non-overlapping peak fluorescent intensities in the plot. Sampled regions are marked with dashed lines in the composite slices. The ratio of reactive cells that showed >5 plaque-like fluorescent spots per cell and/or nuclear enrichment of GFP were expressed as mean ± SD. *n*, total observed cells. *N*, repeats of the assay. Scale bars denote 20 μm. **e** Subcellular localization of empty GFP in *N. benthamiana* epidermal cells. Scale bars denote 20 μm. Significances (*P* < 0.05) in (**b**–**d**) were determined by one-way ANOVA–HSD test (two-sided) and marked by letter symbols. Exact *P* values and details of the one-way ANOVA–HSD test can be found in the Source Data file. Source data are provided as a Source Data file.

it was considered as potential side-effect by the stress from transfect and/or universal localization effect by GFP tag (as is revealed by the empty GFP transfection, Fig. 6e). After heat or drought stress for 60 min, the cytoplasmic ONAC023–GFP proteins were largely relocated into the nucleus (Fig. 6d), supporting the results in rice seedlings and protoplasts (Figs. 4a, 5e). Of note, we observed an intermediate state during stress (45 °C for 35 min and 15% PEG 6000 for 20 min) in which ONAC023–GFP and OsREM1.5–RFP were not only co-localized in the nucleus, but also distributed in plaque-like spots along the plasma membrane (Fig. 6c). These results together supported that the subcellular localization of OsREM1.5 was responsive to stress treatment in coordination with ONAC023.

Next, we investigated the mechanism of stress-induced OsREM1.5 relocation. Previous studies in *Arabidopsis* and *Solanum tuberosum* have hinted regulatory role of phosphorylation-residues within the N-terminal intrinsically disordered regions of group 1 remorins in protein-protein interaction and nanodomain formation[25,26]. As expected, the serine/threonine residues (S80, T92, and S97) of OsREM1.5 are conserved at the reported phosphorylation sites (Fig. 7a). Phos-tag assay with immunoprecipitated OsREM1.5–HA revealed phosphorylation of the protein under normal conditions but not drought or heat stress (Fig. 7b). The phosphoablative OsREM1.5 (OsREM1.5^AAA) with mutations of those residues to Ala resulted in the absence of phosphorylation bands (Fig. 7b), indicating the S80T92S97 site as the primary form of phosphorylated OsREM1.5. Analysis of a public phosphoproteome dataset on drought stressed rice (see Methods) also supported the finding of S80 phosphorylation under normal conditions but not drought stress (Supplemental Fig. 15).

We further investigate the effects of different phosphorylation states on OsREM1.5 by determine the subcellular localization of the phosphoablative OsREM1.5^AAA and phosphomimetic OsREM1.5^DDD (by modifying the S80T92S97 site to Asp) in *N. benthamiana* epidermal cells. OsREM1.5^DDD–RFP exhibited uniform plasma membrane localization under heat or drought stress, and the re-distribution of cotransfected ONAC023–GFP was no longer observed under stress conditions (Fig. 7c). In contrast, OsREM1.5^AAA–RFP exhibited a plaque-like distribution even under 25 °C and showed almost identical distributions to the wild-type OsREM1.5 under heat or drought stress (Fig. 7c). These plaque-like spots reminded us of the nanodomain structures integrated by remorins[26,27]. Subsequently, BiFC assay was performed to investigate if the plaque-like spots formed by dephosphorylated OsREM1.5 are required for OsREM1.5–ONAC023 interaction. The result illustrated that OsREM1.5^AAA interacted with ONAC023 within the nucleus and plasma membrane plaques under heat and drought stress, while OsREM1.5^DDD did not show detectable interaction with ONAC023 (Fig. 7d). BiFC assay in rice protoplasts also confirmed the interaction between the two importin-α proteins and OsREM1.5^AAA but not OsREM1.5^DDD (Supplementary Fig. 14b). Furthermore, overexpression or mutation of *ONAC023* in the *osrem1.5* protoplasts did not alter drought or heat tolerance as revealed by H₂DCFDA (2′,7′-

dichlorofluorescin diacetate) fluorescence staining. Only the *osrem1.5* protoplasts that coexpressed *ONAC023* and *OsREM1.5^AAA* but not *OsREM1.5^DDD* can restore nuclear enrichment of ONAC023 and decrease the reactive oxygen species (ROS) accumulation caused by *OsREM1.5* mutation under stress conditions (Fig. 7e, f). These results together indicate that stress-induced dephosphorylation of the conserved phosphorylation-residues in the N-terminal intrinsically disordered regions of OsREM1.5 (S80, T92, and S97) is essential for cytoplasm-nucleus relocation and the functioning of ONAC023 in response to drought and heat stress.

## Identification of ONAC023 target genes

To unveil the downstream components regulated by ONAC023, RNA-seq and ChIP–seq (using an anti-ONAC023 antibody, Supplementary Fig. 11) analyses were performed on *onac023* and DJ-WT seedlings under drought and heat stresses. 6031 and 3614 DEGs (DJ-WT vs *onac023*) were identified under drought and heat, respectively, based on a unified genome annotation including 28959 active gene models (Fig. 8a, Supplementary Fig. 16, and Supplementary Data 4, 5). Notably, the up-regulated DEGs (4177 under drought stress and 2658 under heat stress) outnumbered the down-regulated DEGs (1854 under drought stress and 956 under heat stress). In addition, the average ChIP–Seq signal ratios around the transcription start sites (TSS) of DEGs indicated that ONAC023 exhibited a higher affinity for the promoter regions of up-regulated DEGs compared to down-regulated DEGs under both treatments (Fig. 8b). Together with the transcriptional activation activity of ONAC023 detected in rice protoplasts (Supplementary Fig. 17), these results suggested that ONAC023 mainly functions as a transcriptional activator under drought and heat stresses. Next, electrophoretic mobility shift assay (EMSA) was performed to investigate the core binding motif of ONAC023, with a 24-bp motif enriched in the ChIP–seq peak regions under both treatments as a starting point (Supplementary Fig. 18a). The result revealed that the conserved NAC binding motif CACG was more crucial for ONAC023 binding than other regions in the 24-bp proto-motif (Supplementary Fig. 18b), indicating that its capability to bind to the same core binding motif as other known stress-responsive NACs, despite belonging to a distinct phylogenetic subgroup[23].

The up-regulated DEGs containing at least one ONAC023-binding site in their promoter regions (−3000 bp to +100 bp in relation to TSS) were determined as putative ONAC023-target genes (ONAC023-TGs) for further analysis. A total of 277 and 1416 ONAC023-TGs were identified under drought and heat stress, respectively (Fig. 8a, c, and Supplementary Data 6). Gene ontology (GO) analysis of the ONAC023-TGs showed significant enrichment on "Oxidation reduction" and "Oxidoreductase activity" under both stresses (Fig. 8d and Supplementary Data 7), suggesting that the participation of ONAC023 in scavenging ROS, which are commonly induced by drought and heat stresses. Nitro blue tetrazolium (NBT) staining showed that the superoxide anion accumulation in *onac023* was significantly higher

than that in DJ-WT after drought or heat treatment (Fig. 8e). This result, together with the H₂DCFDA staining results in rice protoplasts (Fig. 7e, f), suggests that ONAC023 participates in the regulation of ROS homeostasis to fulfill its functions in drought and heat tolerance.

Among the targets associated with oxidative stress tolerance, *PGL3*, an ONAC023-TG under drought stress, was reported to affect ROS accumulation under stress conditions and participate in chlorophyll synthesis[28,29]. These phenotypes were largely in accordance with

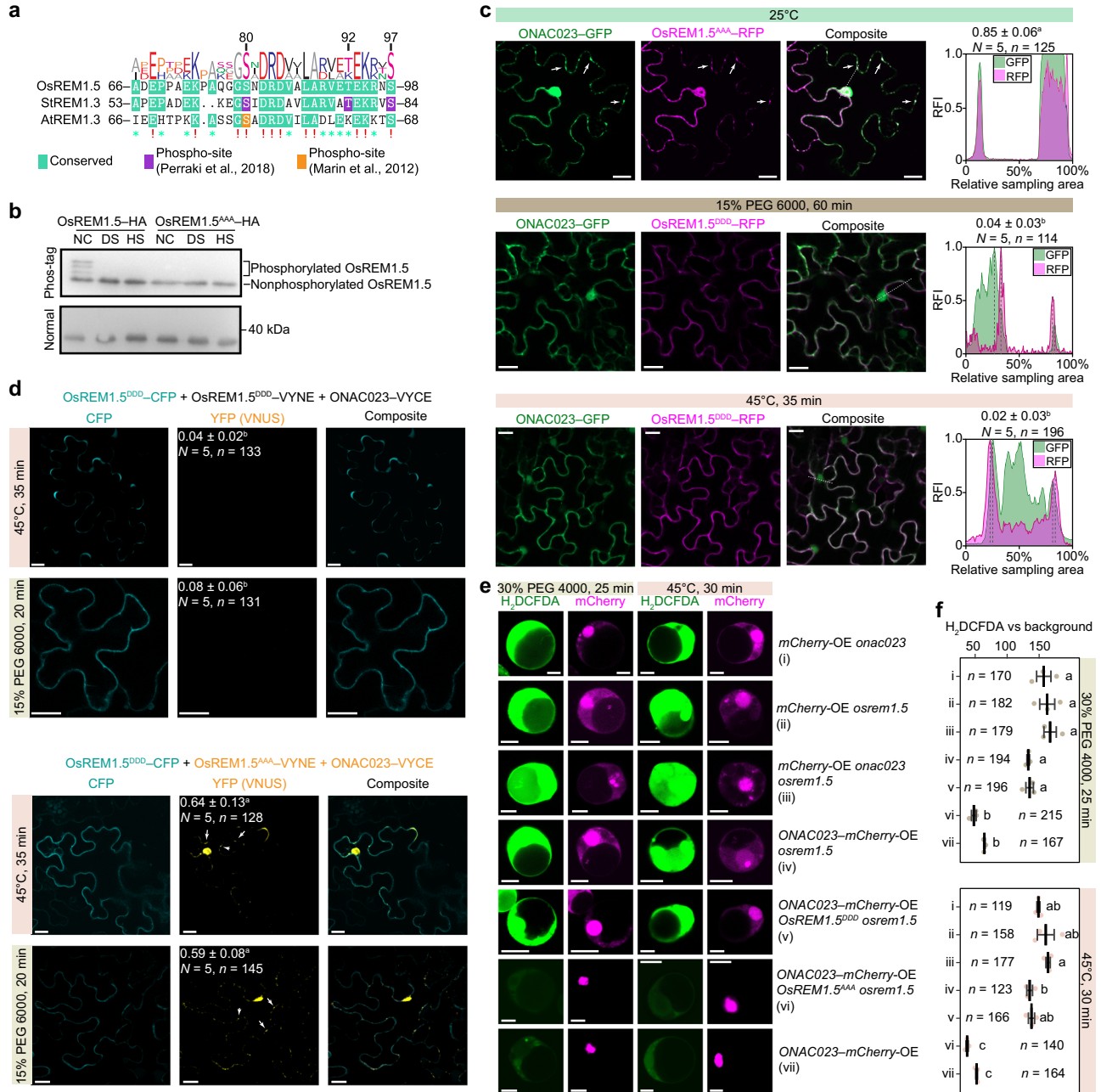

**Fig. 7 | The phosphoablative form of OsREM1.5 contributed to the nuclear translocation of ONAC023 and the decreased cellular ROS accumulation under drought and heat stress. a** Snap-shot of protein sequence alignments of OsREM1.5, AtREM1.3, and StREM1.3 involving the conserved phosphor-residues. **b** Phos-tag assay showing the impaired phosphorylation of OsREM1.5 under drought or heat treatment. The phosphoablative OsREM1.5^AAA was used as a control. NC, normal condition, DS, drought stress, HS, heat stress. See Supplementary Fig. 23 for the uncropped images. **c** Subcellular localization of the phosphomimetic OsREM1.5^DDD and phosphoablative OsREM1.5^AAA under drought (15% PEG 6000) or heat stress (45 °C) and normal conditions (25 °C) in *N. benthamiana* epidermal cells. Typical plaque-like localizations are labeled by arrows. The peak plots illustrate the profiles of the GFP and RFP relative intensities. Differential localization of ONAC023–GFP (cytoplasm) and OsREM1.5^DDD–RFP (cell membrane) was indicated by the non-overlapping dashed lines. The sampled regions are marked with dashed lines in the composite slices. Statistics on the ratio of reactive cells (showed >5 plaque-like fluorescent spots per cell) were denoted as mean ± SD in each panel. *n*, total observed cells. *N*, repeats of the assay. **d** BiFC assay detecting the ONAC023–OsREM1.5^AAA interaction in *N. benthamiana* epidermal cells under drought (15% PEG 6000 for 20 min) or heat stress (45 °C for 35 min). The OsREM1.5^DDD–CFP was used as a transfection control. The ratio of cells showed yellow fluorescence was labeled as mean ± SD in the YFP slices. *n*, total observed cells. *N*, repeats of the assay. **e**, **f** ROS accumulation of the drought (30% PEG 4000) or heat (45 °C) stress-treated transgenic protoplasts identified by H₂DCFDA. Cross-bars in (**f**) denote mean ± SD based on three repeats of the treatments. *n*, total observed cells. Significances (*P* < 0.05) in (**c**, **d**, **f**) were determined by one-way ANOVA–HSD test (two-sided). Exact *P* values and details of the one-way ANOVA–HSD test can be found in the Source Data file. Scale bars denote 20 μm (**c**, **d**) or 10 μm (**e**). Source data are provided as a Source Data file.

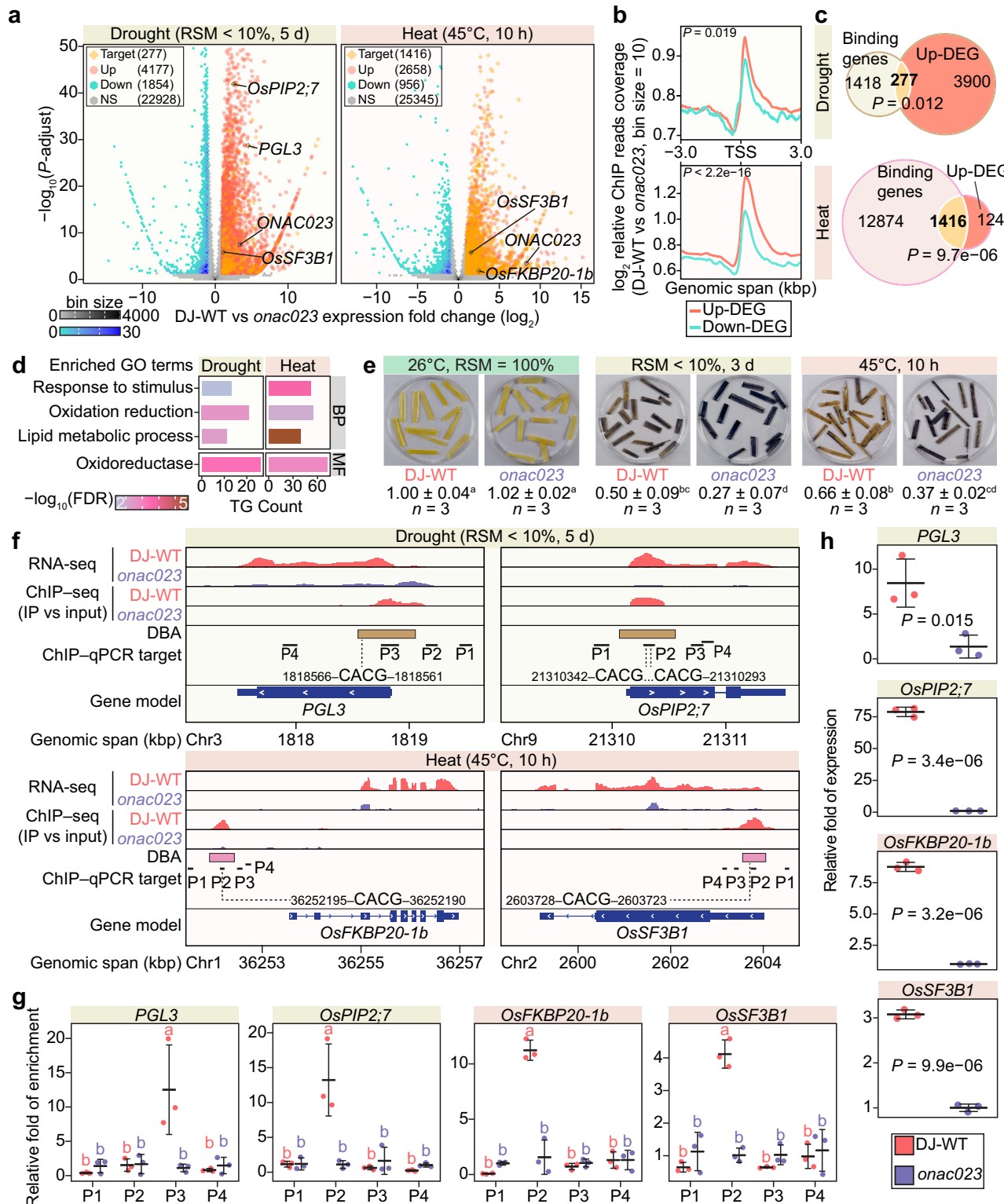

the NBT-staining result (Fig. 8e) and the reduced chlorophyll content in *onac023* under drought stress (Supplementary Fig. 19). Moreover, ONAC023-TGs were reported to function in other stress-responsive mechanisms, such as water transport and alternative splicing regulation. For instance, *OsPIP2;7*, a water channel protein gene facilitating water up-take in leaves through improved water permeability[30], was also identified as an ONAC023-TG under drought stress. This partially explained the lower water content in *onac023* and *ONAC023*-RNAi plants under the drought stress (Supplementary Fig. 5). Interestingly,

ONAC023 also targets genes involved in alternative splicing regulation, such as *OsFKBP20-1b* and *OsSF3B1*, under heat stress. *OsFKBP20-1b*, encoding an immunophilin family protein interacting with the splicing factor OsSR45, regulates alternative splicing in response to environmental stress[31]. *OsSF3B1* encodes a predicted splicing factor 3b subunit, and suppression of *OsSF3B1* led to increased intron retention[32]. The RNA-seq and ChIP–seq landscapes of abovementioned four representative ONAC023-TGs were shown in Fig. 8f, and the interaction between ONAC023 and the CACG core motif in promoter regions

**Fig. 8 | Identification of the ONAC023-TGs under drought stress and heat stress. a** DEGs in terms of wild-type (DJ-WT) vs *onac023* by RNA-seq. Genes investigated in this study are shown in the plot. The *P*-adjust value was determined by Wald test (two-sided) with Benjamini-Hochberg adjustment. **b** Relative ChIP–seq enrichment profiles (DJ-WT vs *onac023*) around the TSS region of the up-regulated DEGs. The enrichment signals were presented as the log$_2$ ratio of DJ-WT vs *onac023* reads per genome coverage, averaged in 10-bp bins. **c** Venn diagrams showing the overlap of ONAC023-binding genes and up-regulated DEGs. The significance of the overlaps was determined by hypergeometric test (two-sided). **d** GO enrichment of ONAC023-TGs. The histogram length represents the enriched TG numbers by the GO term, and the colors represent the significance (by hypergeometric test with Benjamini-Hochberg adjustment, two-sided) of the enrichment. **e** NBT-staining of the *onac023* and DJ-WT leaves. The gray levels under stress were normalized to those under normal conditions and expressed as mean ± SD of three independent

repeats. **f** Track view of the RNA-seq reads intensities and the relative ChIP–seq intensities (IP vs input) of the four representative ONAC023-TGs. The binding areas of ONAC023 are denoted as rectangles. The CACG core binding motifs within the detection ranges (250 bp flanking the amplification region) of the ChIP–qPCR assay were labeled. **g** ChIP–qPCR validation of ONAC023 binding to the promoters of the target genes. Cross-bars indicate mean ± SD of the relative enrichment (IP vs input) determined by three repeats of the treatments (at least 20 seedlings each). **h** Validation of the expression of the four ONAC023-TGs by RT–qPCR. Cross-bars indicate mean ± SD of the relative expression levels for three repeats of the treatments (at least 10 seedlings each). *P* values were determined by two-tailed Student's *t* test in (**b**, **h**). Multi-group comparison significances (*P* < 0.05) in (**e**, **g**) were determined by one-way ANOVA–HSD test (two-sided). Exact *P* values and details of the one-way ANOVA–HSD test can be found in the Source Data file. Source data are provided as a Source Data file.

of four ONAC023-TGs under the corresponding treatment was confirmed by ChIP–qPCR (Fig. 8g). Also, the expression levels of four ONAC023-TGs genes together with six additional DEGs checked by RT–qPCR were consistent with the RNA-seq results (Fig. 8h and Supplementary Fig. 20a, b). Of particular note, a thermo-sensitive male sterility-determining gene *OsAPT2*[33] was also found to be targeted and activated by ONAC023 in inflorescence under heat stress (Supplementary Fig. 20c). As the decreased expression of *OsAPT2* led to male sterility[33], this result can partially explain the essentiality of ONAC023 to pollen fertility under heat stress (Fig. 3c).

### ONAC023 participates in multiple responses through the target genes under drought and heat stresses

To further evaluate the contribution of the ONAC023-TGs to stress tolerance, *OsPIP2;7* was overexpressed in ZH11. During drought stress at the booting stage, two *OsPIP2;7*-overexpressing lines showed significantly increased drought tolerance and higher seed-setting rates compared to ZH11-WT after recovery (Fig. 9a–c). In addition, mutants of *PGL3*, *OsFKBP20-1b*, and *OsSF3B1* were generated by CRISPR-Cas9-mediated mutation and tested for drought and heat tolerance at the seedling stage. The result showed that all three mutants were significantly more sensitive to stress compared to ZH11-WT in terms of survival rate (Fig. 9d). 3,3′-Diaminobenzidine (DAB) staining of the H$_2$O$_2$-treated leaves revealed higher H$_2$O$_2$ accumulation in *pgl3* and *ossf3b1* mutants than in ZH11-WT (Fig. 9e), further supporting the increased H$_2$O$_2$-accumulation in *onac023* under both drought and heat stresses (Fig. 8e). These results together suggest that ONAC023 confers drought and heat tolerance by directly regulating arrays of stress-related genes, thereby integrating multiple stress-responsive processes.

Furthermore, two alternative splicing-related ONAC023-TGs, *OsFKBP20-1b* and *OsSF3B1*, led us to investigate the genome-wide alternative splicing events (ASEs) between *onac023* and DJ-WT. 1928 and 1012 differentially alternatively spliced genes (DASGs) were identified under drought and heat treatments, respectively, using a valid ASE analysis pipeline[34] on the RNA-seq data (Supplementary Fig. 21a–c and Supplementary Data 8, 9). In addition, *ONAC023* mutation led to a decreased percentage of genes with functional protein-coding isoforms revealed by Pfam annotations (Supplementary Fig. 21d, Supplementary Data 10). For instance, ONAC023 was found to promote the production of functional splicing-isoforms, including the annexin gene *OsANN1*, which is known to confer osmotic and heat stress tolerance[35] (Supplementary Fig. 21e and Supplementary Fig. 20c, d). Furthermore, RT–qPCR assays also confirmed the differential ASEs within *OsANN1* in *ossfkbp20-1b* and *ossf3b1* mutants (Fig. 9f, g), suggesting that the absence of ONAC023-targeted alternative splicing regulators (OsFKBP20-1b and OsSF3B1) could cause a possible function loss of *OsANN1* due to the "falsely" retained introns. Our findings provide an example for insight into the intermediary role of ONAC023 in the alternative splicing regulations under abiotic stress. These

results demonstrate that ONAC023 is involved in multiple stress-triggered responses by targeting diverse downstream genes under drought and heat stresses.

## Discussion

In this study, we inferred the multi-stress regulation role of ONAC023, a NAC family member, based on a NAC-centered GRN and genetic data (Fig. 1). Results showed that ONAC023 confers drought and heat tolerance at both seedling and reproductive stages (Figs. 2, 3). Unlike the previously studied stress-responsive NACs from subgroups II and III, ONAC023 phylogenetically belongs to subgroup V, which comprises only two members[23], suggesting that ONAC023 may represent an independent category of stress-responsive NACs with unique regulatory mechanisms.

Our results demonstrated that the activation of ONAC023 function is regulated in two dimensions. The first dimension is the temporal expression induction by stresses. *ONAC023* maintains a rather low or undetectable abundance of transcript and protein under the non-stress conditions (Fig. 1f and Supplementary Fig. 11), but shows an apparent elevation of expression under drought and heat treatments (Fig. 1f), indicating a precisely and timely regulatory response to stress conditions at the transcriptional level. Expression regulation is common among many stress-responsive NACs and this characteristic might have been selected during crop domestication[7,36]. A study on maize *ZmNAC111* identified a drought tolerance-associated 82-bp InDel in its promoter region and the InDel is negatively associated with its expression[36]. Association analysis of *ONAC023* also identified a 76-bp InDel, positively effecting its expression and drought tolerance (Fig. 1d, e). In both cases, the InDels seem to be selected for drought tolerance, implying that the transcriptional regulations on *NAC* genes have been commonly evolved during crop domestication for stress adaptations (Supplementary Fig. 2).

The second dimension of ONAC023 regulation involves subcellular protein translocation in response to drought and heat stresses. Previous studies revealed that the subcellular translocation of certain NACs was regulated at the post-translational level to stringently and rapidly activate their functions as TFs[16,18,37]. The majority of such NACs possess a transmembrane (TM) domain in the proximal C-terminal and undergo truncation before entering the nucleus by peptide cleavage and/or AS, such as ANAC013 and ANAC017 in *Arabidopsis* and OsNTL3, OsNTL5, and ONAC054 in rice[16–18,38,39]. Some of the lipid-associated members were regulated by peptide modifications, such as the *S*-palmitoylation of MfNACsa[40]. However, the nuclear translocation of ONAC023 occurs via a distinct spatial regulation from membrane-bound or lipid-associated NACs. Firstly, ONAC023 does not anchor to the membrane since it lacks a transmembrane domain (Supplementary Fig. 12). Moreover, protein–protein interaction and subcellular localization analyses suggested that ONAC023 may relocate to the nucleus through the remorin–importin-α interaction (Figs. 5–7 and Supplementary Fig. 14). Studies have revealed that group I remorins can form

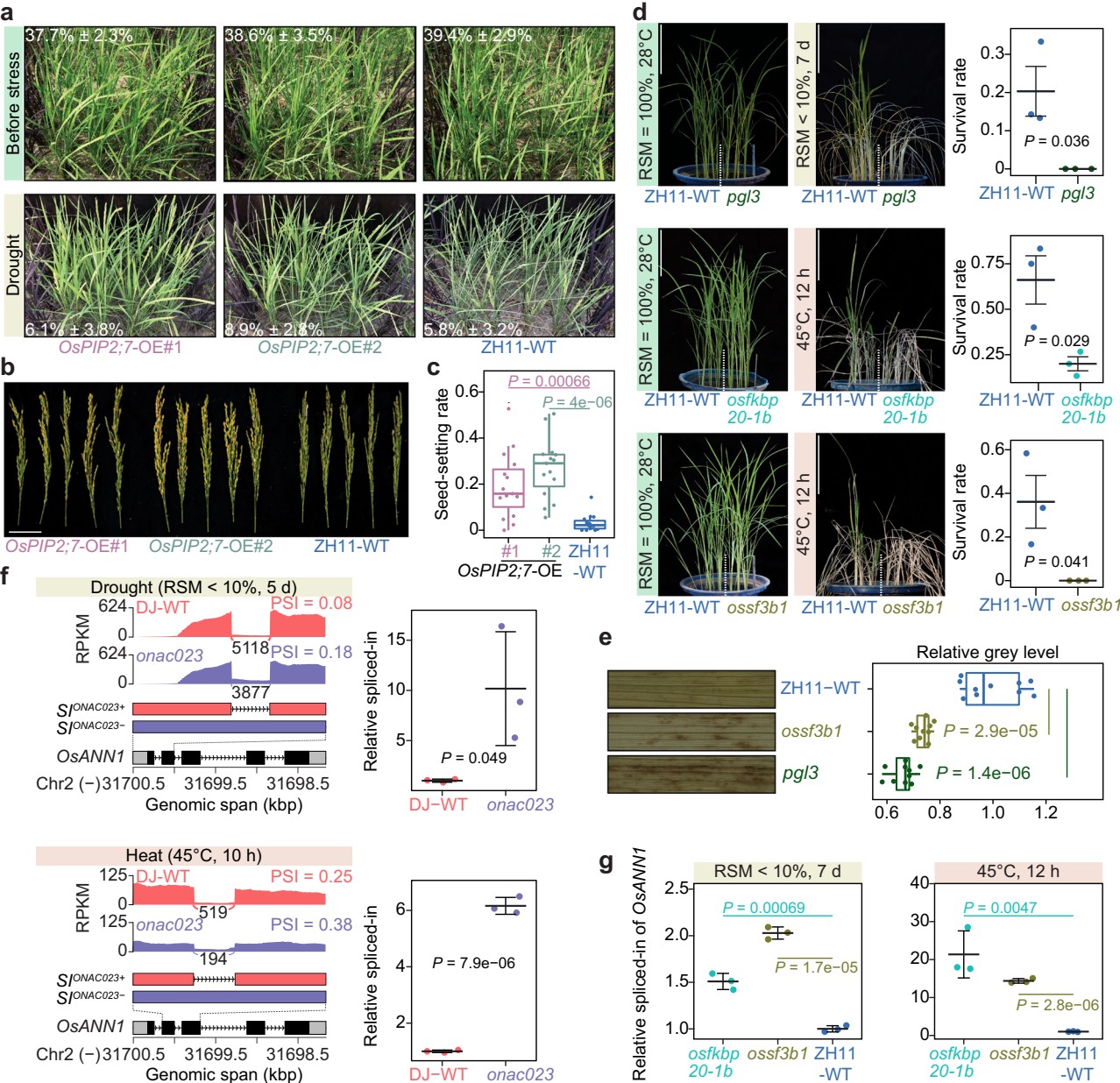

**Fig. 9 | Stress tolerance evaluation of the transgenic lines of the representative TGs. a** Phenotype comparison of *OsPIP2;7*-overexpressing lines (*OsPIP2;7*-OE#1 and *OsPIP2;7*-OE#2) and the negative transgenic control line (ZH11-WT) under normal and drought stress (RSM < 10% for 7 d) conditions in the field. **b** Representative panicles from five plants of the *OsPIP2;7*-OE lines after drought recovery. Scale bar = 5 cm. **c** Seed-setting rates of the *OsPIP2;7*-OE lines after drought recovery. The box plot presents the 1.5× IQR ranges (whiskers) extended from the 25th and 75th percentiles (lower and upper hinges) and median values (central hinges) determined by 15 plants for each tested line in (**a**). **d** Phenotype of the mutant seedlings of the representative ONAC023-TGs before and after drought or heat stress, compared to the ZH11-WT. Scale bars = 10 cm. The cross-bar plots denote the survival rate (mean ± SD) of seedlings from three buckets under independent treatments. **e** DAB-staining result of the 1% $H_2O_2$-treated *pgl3*, *ossf3b1*, and ZH11-WT leaves. Photos show one representative image from ten stained leaf segments each line. The box plot presents the 1.5× IQR ranges (whiskers) extended from the 25th and 75th percentiles (lower and upper hinges) and median values (central hinges) on the relative gray levels (by normalization to ZH11-WT) of the DAB-staining. **f** The sashimi plots indicate the RNA-seq reads densities and the average exon-junction reads counts of the RI-DASEs in the annexin gene *OsANN1* under drought or heat stress. The cross-bars show the relative spliced-in quantifications of the DASEs by RT−qPCR (mean ± SD, data from three repeats of the treatments). **g** The qPCR validations on the *OsANN1* RI-DASEs in the *osfkbp20-1b* and *ossf3b1* mutants under the heat stress treatment. The cross-bars show the relative spliced-in quantifications of the DASEs by RT−qPCR (mean ± SD, data from three repeats of the treatments). Two-tailed Student's *t* test was applied to determine the differences in (**c**–**g**). Source data are provided as a Source Data file.

oligomers through a conserved coiled-coil region in the C-terminal, and these oligomers can further form higher-order nanodomains along the plasma membrane[41]. Both in vitro and in vivo experiments demonstrated that the layout of the nanodomain is controlled by phosphorylation of the N-terminal intrinsic disordered region of remorins, and the dephosphorylated form led to larger domain size[26,27,42]. In agreement with the previous results, phosphoablative

OsREM1.5^AAA showed plaque-like distribution even under normal conditions, while phosphomimetic OsREM1.5^DDD did not form such a structure (Fig. 7c). Therefore, we inferred that the plaques were possibly the aggregation of OsREM1.5-integrated nanodomains. The condensation within the nanodomain often limits protein diffusion and increases dwell time, facilitating protein interactions. This explains why OsREM1.5^AAA, but not OsREM1.5^DDD, showed interactions with

ONAC023 and importin-αs (Fig. 7d and Supplementary Fig. 14b). An earlier study in *Arabidopsis* showed a similar result that phosphor-null mutations of AtREM1.3 promoted its interaction with importins[25], implying a conserved regulatory mechanism for protein translocation in plants.

Our results suggest that the remorin-mediated transcriptional regulation of ONAC023 may occur specifically under stress conditions. This mechanism may also explain why the growth and development of the *ONAC023*-OE plants were not significantly affected under normal conditions (Fig. 2f, Supplementary Fig. 7). Together, these findings highlight *ONAC023* as a valuable gene for stress-resilience engineering without adverse effects caused by overexpression, thus allowing its application in a viable and cost-efficient way.

To achieve a more comprehensive understanding of the regulatory mechanisms of ONAC023, we conducted integrative analyses of the RNA-seq and ChIP–seq data to identify ONAC023-TGs (Fig. 8). Intriguingly, few well-evidenced functional genes in stress signaling pathways or stress tolerance were presented in ONAC023-TGs (Supplementary Data 6), which is distinct to other reported stress tolerance-conferring NACs, such as SNAC1, OsNAC10, NTL3, and OsNAC5[17,43,44]. Instead, ONAC023 targets genes involved in various physiological or molecular processes, including ROS response, water transport, and alternative splicing regulation. Therefore, we propose that ONAC023 confers drought and heat tolerance through integrating these processes. The functional diversification of ONAC023-TGs agrees with the diverse physiological traits or alternative splicing profiles affected by ONAC023 under drought and heat stresses (Fig. 8, Supplementary Figs. 5 and 19). The hypothesis is also supported by the functions of representative ONAC023-TGs such as *PGL3*, *OsPIP2;7*, *OsFKBP20-1b*, and *OsSF3B1*, as well as the performance of their transgenic plants under drought or heat stress (Figs. 8 and 9). These results implicate the sophisticated regulatory mechanisms of ONAC023 and underscore its uniqueness in stress tolerance regulation compared to other reported stress-responsive NACs. In addition, integrated regulation of multiple processes is expected to provide higher robustness when confronting complex environments, which may explain why *ONAC023*-OE lines maintain relatively high yields even under in-field drought and heat stress conditions (Figs. 2d, e and 3d, e, and Supplementary Figs. 6 and 9), a performance not generally reported for most other NACs. To date, the well-known drought and heat stress-responsive mechanisms are relatively independent, with few regulators participating in both stress responses[45]. The broad biological range of ONAC023-TGs suggests that ONAC023 may serve as a hub integrator for cross-talk between drought and heat stress responses. For example, ROS homeostasis, a mechanism crucial for both drought and heat tolerance, represents a convergent point among the processes regulated by ONAC023. The observations of DAB-staining on *ossf3b1* suggest that ONAC023-mediated regulations on alternative splicing may also contribute to ROS homeostasis (Fig. 8e). Further insight into the converged mechanisms underlying ONAC023-TGs may help establish linkages between drought and heat tolerance regulations.

In addition to ONAC023-TGs, we found that alternative splicing for thousands of genes was significantly altered under drought and heat stresses in *onac023* (Supplementary Fig. 21a, b), implying that ONAC023 can somehow affect alternative splicing regulation. Functional prediction on proteins produced by differentially spliced isoforms further revealed that ONAC023 may be involved in producing functional transcripts under stress conditions (Supplementary Fig. 21d, e). Since several genes encoding putative alternative splicing regulators such as OsFKBP20-1b, OsSF3B1, and DEAD-box RNA helicase[31,32,45] are targeted or transcriptionally affected by ONAC023 (Fig. 8f, g, Supplementary Fig. 20a, b, Supplementary Data 6, 8), we speculate that ONAC023 may fine-tune the landscape of alternative splicing through transcriptional regulation of these alternative splicing

regulatory genes. It is noteworthy that the gene composition and predicted functions of DEGs/TGs and DASGs are largely distinct (Fig. 8d and Supplementary Fig. 21c), suggesting relatively independent processes for ONAC023-mediated transcriptional and post-transcriptional alternative splicing regulation. A similar situation was reported in the mineral homeostasis processes in rice[34]. Although the aforementioned studies hypothesized that transcriptional and alternative splicing processes may cooperate to regulate environmental adaptations[45], the key factor coupling these two processes is unknown. Our data indicated that ONAC023 not only regulates a large number of genes involved in various stress response processes at the transcriptional level, but also modulates transcript splicing outputs under drought and heat stresses, thus providing a candidate for the direct coordination between transcriptional and alternative splicing regulations.

By integrating the elucidated regulatory roles, we propose a working model for ONAC023 in the regulation of drought and heat tolerance (Fig. 10). Under normal conditions, *ONAC023* maintains low expression, and the ONAC023 proteins, if any, are retained mainly in the cytoplasm, thereby impeding its stress-specific regulatory function. Upon drought or heat stress stimulation, the expressions of *ONAC023* and *OsREM1.5* are induced, resulting in the removal of phosphorylation in the N-terminal intrinsic disorder region of OsREM1.5. This modification facilitates interactions between OsREM1.5 and ONAC023/importin-αs, enabling the relocation of ONAC023 to the nucleus with the assistance of importin-αs. The nuclear ONAC023 facilitates stress response regulation principally by targeting and activating various stress-related genes functioning in diverse physiological processes such as ROS scavenging and water transport. Besides, ONAC023 mediates alternative splicing regulation to some extent through activating alternative splicing regulatory factors, thereby resulting in a higher proportion of isoforms encoding functional proteins responsible for stress responses or tolerance. Together with the genetic evidence of *ONAC023* in improving drought and heat tolerance, we propose that ONAC023 is not only a valuable model for uncovering the multidimensional regulatory mechanisms of hub regulators in multi-stress responses, but also a promising candidate for drought and heat tolerance breeding in rice.

## Methods
### Plasmid construction and genetic transformation
To create the OE constructs, the coding-sequence (CDS) of *ONAC023* or *OsPIP2;7* was amplified and inserted into the pU1301 backbone (driven by a maize *polyubiquitin-1* promoter)[46]. The *ONAC023* CDS tagged with a 3× FLAG sequence at the 3' end was integrated into pU1301 to generate the *ONAC023–FLAG*-OE construct. The *ONAC023*-RNAi construct was generated by insertion of the palindromic sequence of a partial *ONAC023* CDS fragment into the pDS1301 vector[47]. The above constructs were introduced into ZH11 calli through *Agrobacterium*-mediated transformation[48]. The gene body and the 3500-bp upstream genomic sequence of *ONAC023* were amplified from DJ genomic DNA and integrated into the pCAMBIA2301 backbone to generate the complementary construct, which was subsequently used to transfect the *onac023*-derived calli. The CRISPR-Cas9-mediated mutation construct of *OsREM1.5* was generated based on two guide RNAs using the pRGEB32 vector[49]. Primers used in this study are listed in Supplementary Data 11.

### Plant materials and abiotic stress treatments
All the genetic materials, except *onac023* and *onac023 ONAC023*, were generated in the ZH11 background. The DJ background T-DNA insertion mutant *onac023* (accession ID: 2D-30113) was obtained from Postech (S. Korea, Plant Functional Genomics Laboratory)[50], and the corresponding CK (DJ-WT) was wild-type plants derived from the heterozygous $T_1$ generation of *onac023*. The expression of *ONAC023*

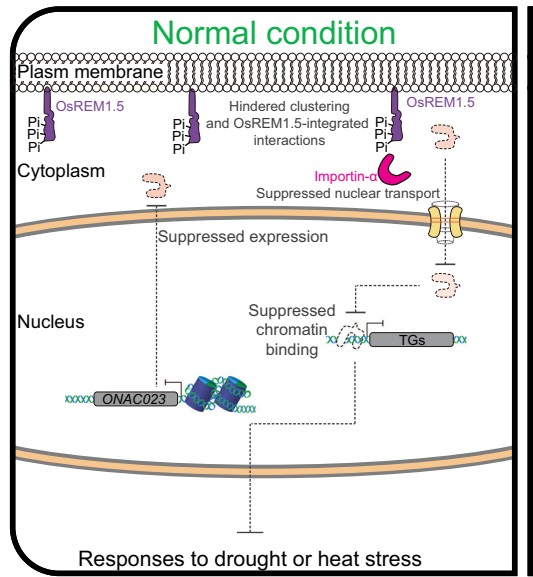
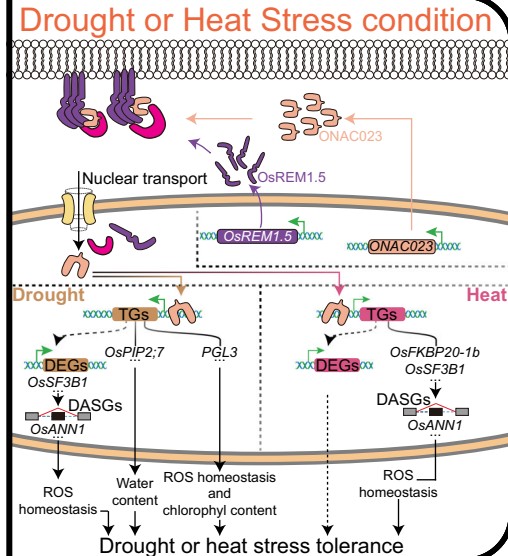

**Fig. 10 | Proposed working model of ONAC023 in conferring drought and heat tolerance.** Under the non-stress condition (the left part), the expression of *ONAC023* is suppressed, and the minimal (if exist) ONAC023 proteins are tethered in the cytoplasm with the absence of OsREM1.5-integrated interactions, thereby impeding its stress-specific regulatory function. Under drought or heat stress (the right part), the up-regulated expression of *OsREM1.5* and the dephosphorylation of the N-terminal intrinsically disordered region of OsREM1.5 confer enhanced interaction with ONAC023 and importin-αs. Then ONAC023 is transported to the nucleus possibly with the help of Importin-αs. The nuclear ONAC023 regulates stress response by targeting and activating arrays of stress-related genes functioning in diverse processes such as ROS scavenging, water transport, and alternative splicing regulation.

was detected by RT–qPCR and the copy numbers were analyzed by genomic DNA-based qPCR (Supplementary Fig. 22)[51,52]. Two single-copy transgenic lines from each construct were selected for further analysis, and the corresponding CK plants (ZH11-WT) were derived from the $T_1$ heterozygous transgenic plants. Seeds from two mutant-complementation lines with similar drought-induced *ONAC023* expression and plant morphology were combined equally and used as one transgenic line (*onac023 ONAC023*) in the experiments. The CRISPR-Cas9-mediated mutants of the ONAC023-TGs were generated by BIOGEL GeneTech Co., Ltd (Wuhan, China) with the accession numbers #1702-26560 (*osfkbp20-1b*), #1702-26562 (*ossf3b1*), and #1702-26564 (*pgl3*). $T_3$ (or higher) generation transgenic lines with stable expression levels and phenotypes were used in the experiments. All CRISPR-Cas9-mediated mutants used in the phenotyping are Cas9-free, and the target sites were checked by sequencing (Supplementary Fig. 22e).

For in-field tests, at least three randomized blocks (regarded as biological replicates) were planted in Wuhan, Hubei province (30°28′15.84″ N, 114°21′6.42″ E) or Lingshui, Hainan province (18°32′34″ N, 110°2′7″ E). The drought stress was performed by withdrawing at the booting stage. The relative soil moisture (RSM) at 20 cm below the ground surface was recorded using TRIME-PICO32 (IMKO Micro-modultechnik GmbH) with 40% as a critical point of drought stress. After drought stress for 10–16 d, irrigation was supplied for 14–20 d followed by trait evaluation. Drought tolerance-related i-traits were evaluated on the high-throughput phenotyping platform[20]. For each line, three healthy 30-d-old plants planted in buckets at booting stage were subjected to drought stress. After the water supply was cut off, RSM was measured and adjusted to an equal value until it reached 40%, followed by 360° images scanning of the plants at 8:00 am everyday during the 5-d drought stress. GPAR-R was calculated based on the percentage of green areas in the images (the value of day 0 was set to 1). For in-field heat stress, blocks with the flowering time overlapped with the high temperature period (peak air temperature exceeds 35 °C at least for 3 d) were selected for yield trait evaluation. Two-year replicates were performed for in-field heat stress that occurred in August in Wuhan (see Supplementary Fig. 7 for the air temperature records during heading and flowering).

For seedling drought stress treatment, water was withheld for the 4-leaf stage seedlings planted in sandy soil (15 seedlings for each line per replicate) until the RSM was below 10% for 5–7 d, and the survived seedlings were recorded after rewatering for 10 d. For heat stress at the seedling stage, 4-leaf stage seedlings grown in the green house with a 25–28 °C air temperature were moved to a growth chamber with the air temperature of 45 °C, relative humidity of 90%, and a light intensity of 15000 lux. After treatment for 10–12 h, plants were moved back to the green house for 7–14 d recovery. Each treatment was repeated at least three times.

For in-house heat stress treatment at the flowering stage, three plants per replicate were kept in a growth chamber for 3–5 d at 43 °C for 4 h at noon. Spikelets flowered during the heat stress treatment were sampled and used for pollen germination assays[24]. Seed-setting rates of the recovered plants were measured after grain-filling the recovered plants. Each treatment was repeated at least three times.

The central 5 cm fragments of the leaves at booting stage were used for physiological measurements. Five plants were sampled for each of the three (six for chlorophyll content measurement) biological replicates. Stress-related physiological indexes, including free proline content in leaves[53], DAB-staining and NBT-staining-mediated ROS detection, chlorophyll contents, relative water content (RWC)[54], and relative electrolyte leakage (REL)[55] were measured.

**Identification of ONAC023-interacting proteins**

For IP-MS, *ONAC023–FLAG*-OE#5 and ZH11-WT (control) seedlings under RSM < 10% for 3 d (drought stress) or 45 °C for 8 h (heat stress, 30 plants from each of the two repeats of the treatments) were harvested and crosslinked in 0.1% EGS for 20 min followed by 1% formaldehyde for 20 min on ice in a vacuum chamber. Proteins were extracted using IP buffer (50 mM HEPES, pH 8.0, 150 mM NaCl, 5 mM $MgCl_2$, 10% glycerol, 0.1% Triton X-100, 1 mM DTT, 1 mM PMSF, and protease inhibitor cocktail [Roche, 4693132001, one tablet for 50 mL]) from 1 g tissue and incubated with anti-FLAG magnetic beads (Sigma-

Aldrich, M8823) overnight at 4 °C. After being washed with the IP buffer six times, proteins were eluted by incubation with the SDS–PAGE sample buffer for 15 min at 100 °C and briefly separated in the SDS–PAGE gel. The gel regions containing proteins were cut and destained in 50 mM NH₄HCO₃ in 50% acetonitrile. Then the gel pieces were dehydrated with 100 μL of 100% acetonitrile for 5 min, then rehydrated in 10 mM dithiothreitol and incubated at 37 °C for 60 min. After that, gel pieces were again dehydrated in 100% acetonitrile before rehydration with 55 mM iodoacetamide. Then samples were incubated at room temperature in the dark for 45 min. Next, the gel pieces were washed with 50 mM NH₄HCO₃ and dehydrated in 100% acetonitrile. After that, gel pieces were resuspended in 50 mM NH₄HCO₃ with 10 ng/μL trypsin (New England BioLabs, P8101S) on ice for 1 h. After the excess liquid was removed, the gel pieces were placed at 37 °C for overnight trypsin digestion. Peptides were extracted with 50% acetonitrile/5% formic acid, followed by 100% acetonitrile. Peptides were freeze-dried and resuspended in 2% acetonitrile/0.1% formic (used as solvent A). Liquid chromatography with tandem mass spectrometry (LC-MS/MS) was performed by PTM Biolabs Inc. (Hangzhou, China). In detail, liquid chromatography was performed using an EASY-nLC 1000 UPLC system (Thermo Fisher). The solvent A was directly loaded onto a home-made reversed-phase analytical column (15 cm length, 75 μm inner diameter). The gradient consisted of an increase from 5% to 25% solvent B (0.1% formic acid in 90% acetonitrile) over 11.5 min, 25% to 35% over 1.5 min and an increase to 80% over 1 min, before maintenance at 80% for the last 1 min, all at a constant flow rate of 1000 nL/min. The peptides were subjected to nano spray ion source followed by tandem mass spectrometry (MS/MS) with the Q Exactive™ Plus system (Thermo Fisher) coupled online to the UPLC. The electrospray voltage was 2.2 kV and the m/z scan range was 350–1800 for full scan. Intact peptides were detected in the Orbitrap at a resolution of 70,000. Peptides were then selected for MS/MS with a normalized collision energy of 28%, and the fragments were detected in the Orbitrap at a resolution of 17,500. A data-dependent acquisition procedure that alternated between one MS scan followed by 20 MS/MS scans with 15.0 s dynamic exclusion was used. The automatic gain control was set to 5e4[56]. The output files of MS/MS database search using the software MS-GF+ against the MSU v7.0 peptide dataset with default parameters[57] were decoded and filtered with the R package MsnID [https://bioconductor.org/packages/release/bioc/html/MSnID.html], and the peptides with an E-value < 1e − 05 were preserved. The proteins containing peptides presented in both replicates of ONAC023–FLAG-OE#5 but not in ZH11-WT were regarded as ONAC023-interacting proteins.

The full-length CDSs of ONAC023, SNAC1, OsREM1.5, and the 1–240 bp truncation of OsREM1.5 CDS were cloned into pVYNE or pYVCE[58] for BiFC assay in rice protoplasts, and the fusions of these CDSs with an N-terminal HA or FLAG tag were cloned into pU1301 for Co-IP assay. ZH11 protoplasts were prepared using 14-d-old seedlings growing on half-strength Murashige and Skoog media. Seedling sheath strips were digested in the digestion solution (10 mM MES, pH 5.7, 0.5 M mannitol, 1 mM CaCl₂, 5 mM β-mercaptoethanol, 0.1% BSA, 1.5% cellulase R10, and 0.75% macerozyme R10) for 4 h at 25 °C in the dark. The collected protoplasts were incubated in W5 solution (2 mM MES, pH 5.7, 154 mM NaCl, 5 mM KCl, and 125 mM CaCl₂) at 25 °C for 60 min. Then, the protoplasts were collected by centrifugation at 100 g for 5 min and resuspended in MMG solution (4 mM MES, pH 5.7, 0.6 M mannitol, and 15 mM MgCl₂) to a final concentration of ~1.0 × 10⁷ mL⁻¹. After that, ~3 μg plasmid was gently mixed with 100 μL protoplasts and 110 μL PEG-CaCl₂ solution (0.6 M mannitol, 100 mM CaCl₂, and 40% PEG4000) and incubated at room temperature for 10 min. 440 μL of W5 solution was then added to stop the transformation, and the protoplasts were collected by centrifugation, resuspended in WI solution (4 mM MES, pH 5.7, 0.6 M mannitol, and 4 mM KCl), and cultured in the dark. Cells for BiFC were collected after 16-h dark incubation and

examined by laser scanning confocal microscopy (Olympus, FV1200) for YFP imaging. The BiFC fluorescent cell ratio was recorded based on observations of >100 cells in each of the five repeats of the assay. For BiFC assay in N. benthamiana epidermal cells, CDSs of ONAC023 and the phosphoablative/phosphomimetic mutants of OsREM1.5 were cloned into pVYCE and pVYNE and transformed into the Agrobacteria GV3101 strain for infiltration of 5–6-week-old N. benthamiana leaves.

For Co-IP assay, protoplasts were cultured for 24 h at 25 °C, and proteins from ~10⁷ cells were extracted using 2 mL IP buffer. 10% of the extract was preserved as input, and the rest was incubated with Anti-HA Affinity Matrix (Roche, 11815016001) or anti-FLAG magnetic beads (Sigma-Aldrich, M8823) overnight at 4 °C. The beads were washed six times before elution by incubation with SDS–PAGE sample buffer for 10 min at 100 °C. The eluted protein and input samples were electrophoresed on an SDS–PAGE gel and transferred to a PVDF membrane for Western blot. The membrane was blocked in 1×PBST containing 5% nonfat milk and 0.1% Tween 20 for 1 h and incubated with anti-FLAG (Sigma-Aldrich, F3165, 1:5000 dilution) or anti-HA (ABclonal, AE008, 1:2500 dilution) antibodies overnight at 4 °C. The membrane was washed in PBST three times and incubated with the secondary antibodies (1:10,000 dilution) for 1 h. Then the membrane was washed four times in PBST before band detection by Clarity Western ECL Substrate (Bio-Rad, 1705061) and imaging by ChemiDoc XRS+ imaging system (Bio-Rad, 1708265). See Supplementary Fig. 23 for uncropped images with molecular weight markers.

For in vitro pull-down assays, ONAC023 and OsREM1.5 CDS were cloned to pGEX-4T-1 (with GST tag) and pET-28b-SUMO (with 6× His tag), respectively, and then expressed using E. coli strain BL21. Proteins were purified after 12-h incubation at 30 °C, 200 rpm. Purified proteins were pooled with equal concentration in binding buffer (20 mM Tris–HCl, pH 8.0, 200 mM NaCl, 0.5% IGEPAL CA-630, 1 mM EDTA, and protease inhibitor cocktail [Roche, 4693132001, one tablet for 50 mL]) and incubated for 4 h at room temperature. After 3-h incubation with Glutathione Sepharose beads (GE Healthcare, 17075601) at 4 °C followed by washing with binding buffer five times, proteins were eluted by adding 100 μL SDS–PAGE sample buffer and incubating at 100 °C for 10 min. The eluted protein and corresponding input proteins (fractions not incubated with the beads) analyzed by Western blot with anti-His (Abclonal, AE003, 1:3000 dilution) and anti-GTS (Abclonal, AE001, 1:5000 dilution) antibodies. An irrelevant protein, OsPYL1–SUMO-6× His, was used as a control.

## Subcellular localization identification

For immunofluorescence assay, leaves from ONAC023–FLAG-OE#5 and ZH11-WT 4-leaf stage seedlings under drought (RSM < 10% for 3 d), heat (45 °C for 3 h), and normal (25 °C with ample water) conditions were sampled and fixed for 30 min in 4% paraformaldehyde and then embedded in 5% agarose with a 100-μm-thick longitudinal vibration section. The samples were blocked by PBST with 5% nonfat milk and subsequently incubated with an anti-FLAG antibody (Sigma-Aldrich, F3165) followed by an Alexa Fluor 594-conjugated secondary antibody (Abclonal, AS054). The slices were stained with DAPI (4′,6-diamidino-2-phenylindole) before they were imaged by laser scanning confocal microscopy (Olympus, FV1200).

To determine ONAC023 subcellular localization in ZH11 protoplasts, ONAC023 CDS was cloned into the pM999-YFP vector to generate an ONAC023–YFP construct. Two guide RNAs targeting OsREM1.5 CDS were inserted into pRGEB32 and co-transfected with the ONAC023–YFP construct to generate mutations in OsREM1.5. An empty pM999-YFP vector was transfected as a control. The Ghd7–CFP construct was co-transfected as a nuclear marker[59]. Transfected protoplasts were incubated for 36 h before imaging. Four aliquots of the cells were harvested for genotyping osrem1.5 (Supplementary Fig. 12). For fluorescence co-localization analysis, the YFP and CFP fluorescent intensities of the pixels adjacent to the nucleus in each observed cell

were measured by the "Plot Profile" tool in ImageJ [https://imagej.net]. The intensity values for each fluorescence were normalized by the Z-transform for each cell so that pixels from different cells could be pooled for PCC calculation. At least 12 cells for each transfection were used in the measurement. For subcellular localization detection in *N. benthamiana* epidermal cells, CDSs of *ONAC023* and *OsREM1.5* were cloned into the pCAMBIA1301S-GFP and pCAMBIA1301S-RFP vectors, respectively, and transfected *N. benthamiana* leaves as described in the BiFC assay.

For Western blot, cytoplasmic and nuclear proteins from 12 seedlings for each treatment were separated by the same method for nucleus isolation in the ChIP−seq assay. The supernatant extracted by EB1 buffer was used as the cytoplasmic fraction, and the nuclear extract purified by EB3 buffer was used as the nuclear fraction. Western blot was performed as described in the Co-IP assay. Cytoplasmic and nuclear protein concentrations were adjusted according to anti-Actin (Abclonal, AC009) and anti-histone H3 (Abclonal, A2348) bands, respectively, and equal amounts of samples were loaded for all the blots. ONAC023−FLAG proteins in each fraction were detected by anti-FLAG antibody (Sigma-Aldrich, F3165).

### ROS detection in rice protoplasts

To construct *ONAC023−mCherry-OE OsREM1.5$^{AAA}$/OsREM1.5$^{DDD}$ osrem1.5* protoplasts, the −2000-bp genomic region (proximal promoter) of OsREM1.5 was fused with the full-length CDS of *OsREM1.5$^{AAA}$* or *OsREM1.5$^{DDD}$* and cloned into the pCAMBIA2301 that carrying a *Pro$_{ubi}$::ONAC023−mcherry* insertion. Then the vector was transferred into the *osrem1.5* protoplasts. The *onac023 osrem1.5* protoplasts were constructed by transferring the *OsREM1.5*-pRGEB32 vector into the protoplasts derived from *onac023*. After 24-h incubation in the dark at 25 °C, the transfected protoplasts were treated with 30% PEG 4000 for 25 min (a simulation for drought stress) or 45 °C for 30 min. After that, 10 μM H$_2$DCFDA (Sigma-Aldrich, D6883) was applied to the protoplasts and incubated for 15 min. Then the protoplasts were wash three times with W5 solution before confocal imaging with excitation at 488 nm and emission at 505−525 nm by a laser scanning confocal microscopy (Olympus, FV1200). The microscopy settings were kept stable for all transfects. The ROS accumulation was quantified by the ratio of the mean H$_2$DCFDA intensity of the observed cells vs the mean background intensity of the slice. The assay was repeated three times and > 100 cells for each transfection were involved in the quantification.

### OsREM1.5 phosphorylation detection

The CDS of *OsREM1.5* or *OsREM1.5$^{AAA}$* were cloned into pCAMBIA1301S-HA vector and transfected *N. benthamiana* leaves as described in the BiFC assay. Protein of the transfected leaves were harvested using the phosphorylation preservation buffer (20 mM Tris−HCl, pH 8.0, 200 mM NaCl, 10% glycerol, 0.5% Triton X-100, 1 mM PMSF, 1 mM NaF, 5 mM β-glycerophosphate, and protease/phosphatase inhibitor cocktail [Roche, 4693132001 and 4906845001, one tablet each for 50 mL]) after the drought (15% PEG 6000 for 20 min) or heat (45 °C for 35 min) stress treatment. IP of the HA-tagged protein were performed as described in Co-IP with additional wash using Triton X-100 and NaCl-free extraction buffer. The eluted proteins were loaded with 1 mM MnCl$_2$-containing sample buffer and separated in Phos-tag gel (10% SDS−PAGE with 20 μM Phos-tag [Wako, 300-93523] and 40 μM MnCl$_2$) at 4 °C in the dark. After that, the gel was immersed in 10 mM EDTA-containing transfer buffer thrice with gentle agitation, and then immersed in EDTA-free transfer buffer for 30 min before transferred to a PVDF membrane for Western blot.

### RNA extraction and expression level quantification

Total RNA was extracted from the 14-d seedlings treated by drought (RSM < 10% for 5 d) or heat (45 °C for 10 h) stress using TRIzol reagent (Invitrogen, 15596018), and reverse-transcribed by SuperScript reverse transcriptase (Invitrogen, 18080093) after the removal of genomic DNA by DNase I (Invitrogen, 18068015). RT−qPCR was performed using SYBR Premix ExTaq (TaKaRa, RR81LR) in a ViiA 7 Real-Time PCR System (Applied Biosystems). The relative quantification was calculated using the 2$^{-ΔΔCt}$ method[60]. The rice *Ubiquitin* gene was used as an internal reference. For alternative splicing event quantification, the relative spliced-in of the variable exon was quantified against the total transcript. At least 10 individuals were sampled for each of the three independent repeats of the stress treatments, and 2−3 technical replicates were tested for each sample.

For RNA-seq library construction, 2 μg of total RNA was used to produce the libraries using TruSeq® Stranded Total RNA Library Prep Plant (Illumina, 20020610). The libraries were sequenced by NovaSeq 6000 platform (Illumina). At least 10 seedlings from each of the three repeats of the treatments were sequenced for each line.

### RNA-seq data and alternative splicing analysis

Raw RNA-seq reads were filtered by fastp (-q 30)[61] and aligned to the rice genome assembly MSU v7.0 [http://rice.uga.edu] by SATR v2.52b[62]. Transcripts were assembled from uniquely mapped read pairs by StringTie v1.3.1 in reference guided mode based on non-redundant gene models from MSU v7.0 and IC4R to construct the unified reference transcripts. Transcripts with a TPM larger than 1 in at least one sample were used as the unified reference transcripts for all the subsequent sequencing analyses in this study (Supplementary Data 4). The peptide-encoding capacity of the transcripts in the unified reference was assessed by CPC2[63], and the CDS and corresponding peptide sequences were characterized by CodAn[64].

DEG analysis were performed using R package DESeq2[65]. Genes with more than twofold changes and a *P*-adjust < 0.05 (Wald test with Benjamini-Hochberg correction) were defined as DEGs. ASE and DASG analyses were performed by rMATS[34,66].

For functional annotation of DEGs and DASGs, GO enrichment analysis was performed by AgriGO v2.0[67]. Predicted functions of the DASG-encoding peptides were annotated with HMMER v3.3.2[68] against the Pfam database v32.0 [http://pfam.xfam.org]. Only the Pfam annotation with the lowest *E*-value (<1e − 05) was considered to represent the protein-encoding functionality of the SI$^{ONAC023+}$ and/or SI$^{ONAC023−}$ (splicing isoform induced/suppressed by ONAC023) for a DASG. The quality control of RNA-seq is shown in Supplementary Fig. 16e−g.

### ChIP-coupled qPCR and high-throughput sequencing

Samples used for ChIP−seq were collected from the same batch of seedlings for the RNA-seq. ChIP was performed using a proved protocol in rice[69]. In detail, 2 g seedlings were crosslinked using 1% formaldehyde and quenched with glycine. Then, the samples were ground in liquid nitrogen into fine powder and resuspended in 30 ml of buffer EB1 (0.4 M sucrose, 10 mM Tris−HCl, pH 8.0, 5 mM β-mercaptoethanol, 1 mM PMSF, and Protease Inhibitor cocktail). The mixture was filtered through Miracloth (Millipore, 475855), and the filtrate was centrifuged at 1800× *g* for 10 min at 4 °C. The pellet was resuspended in 1.5 ml of buffer EB2 (0.25 M sucrose, 10 mM Tris−HCl, pH 8.0, 5 mM β-mercaptoethanol, 10 mM MgCl$_2$, 1% Triton X-100, 1 mM PMSF, and Protease Inhibitor cocktail) and centrifuged at 2000× *g* for 10 min at 4 °C. Next, the pellet was resuspended in 0.5 ml of buffer EB3 (1.7 M sucrose, 10 mM Tris−HCl, pH 8.0, 5 mM β-mercaptoethanol, 2 mM MgCl$_2$, 0.15% Triton X-100, 1 mM PMSF, and Protease Inhibitor cocktail) and centrifuged at 2000× *g* for 1 h at 4 °C. The final pellet was resuspended in 0.25 ml of buffer NLB (10 mM Tris−HCl, pH 8.0, 20 mM EDTA, 400 mM NaCl, 1% Triton X-100, 2 mM PMSF, and Protease inhibitor cocktail), and the chromatin was fragmented into 200−600 bp by sonication. DNA was immunoprecipitated by overnight incubation with Dynabeads protein A (Invitrogen, 10001D)-conjugated anti-ONAC023 polyclonal antibody (Abmart, 8114-1hz, validated in Supplementary Fig. 11) for DJ-

WT and *onac023* samples, and anti-FLAG magnetic beads (Sigma-Aldrich, M8823) for *ONAC023–FLAG*-OE#5 and ZH11-WT.

ChIP-coupled quantitative PCR (ChIP−qPCR) was performed using the fold enrichment method, in which $\Delta C_t = C_t^{IP} - C_t^{input}$. For all ChIP−qPCR experiments, stress treatments were repeated three times (each with at least 10 plants), and 2–3 technical replicates were performed for each reaction. ChIP−seq libraries and control libraries based on input DNA (sonicated chromatin) were prepared by NEB-Next® Ultra™ II DNA Library Prep Kit for Illumina® (New England Bio-Labs, E7645) following the manufacturer's instructions. All libraries were sequenced using NovaSeq 6000 platform (Illumina). At least two biological replicates were sequenced for each line.

## ChIP−seq data analysis
The raw sequencing reads were filtered by fastp[61] and mapped to rice genome assembly MSU v7.0 by bowtie2[70] with parameters "--sensitive, --no-unal, -X 1000". Peak calling was performed on nonredundant, uniquely mapped reads by MACS2[71]. Peaks called in at least two replicates were applied to DBA (differential-binding area) analysis by R package DiffBind[72]. Peaks with more than twofold changes and FDR < 0.05 were defined as DBAs and their associated genomic features were annotated by R package ChIPseeker[73]. Genes with DBAs in their promoter regions (the −3000 to +100 regions in relation to TSS) that enriched in DJ-WT compared to *onac023* were identified as ONAC023-binding genes. MEME program was used for identifying consensus ONAC023-binding sequences[74], and the sequence with an E-value < 1 e − 10 and central enrichment pattern in the peak region revealed by Centrimo[75] was considered as a candidate ONAC023-binding motif (Supplementary Fig. 18). The quality control information for ChIP−seq is shown in Supplementary Fig. 16a–d.

## Electrophoretic mobility shift assays
GST-fused ONAC023 was prepared as described in in vitro pull-down assay. The 5′ FAM-labeled probe was synthesized by Tsingke Bio-technology Co., Ltd. (Beijing, China). To generate double-stranded oligos, an equal amount of the complementary single-stranded oligos was mixed, heated to 95 °C for 2 min, and annealed by gradually cooling down to 25 °C. EMSA was performed using LightShift Chemiluminescent EMSA Kit (Thermo Fisher, 20148). For the competition assay, nonlabelled probes with or without mutated base pairs were incubated with ONAC023–GST in binding buffer at room temperature for 20 min. After that, 1 μL 5′ FAM-labeled probe (10 μM) was added and incubated at room temperature in dark for 20 min. The samples were then subjected to electrophoresis on 6% PAGE gels (SDS-free) with 0.5×TBE buffer at 4 °C in dark for 1 h. The fluorescence signaling was captured by a FLA-5100 scanner (Fujifilm).

## Dual-LUC assay
Full-length and truncated fragments of *ONAC023* CDS were inserted into the GAL4BD vector and then transfected into ZH11 protoplasts together with the reporter vector *ProGAL4:fLUC* and the reference vector *ProAtUbi:rLUC* (see Supplementary Fig. 17)[76]. The luciferase activity was measured using cells cultured overnight with Dual-Luciferase Reporter Assay System (Promega, E1910), and expressed as the chemiluminescence intensity ratio of fLUC vs rLUC. The assay was repeated three times and three independent transfection batches were assayed each time with three technical replicates for each reaction.

## Key stress-related NACs identification by integrative analysis
Key stress-related NACs were identified in three steps. Firstly, the GRN was constructed to evaluate the transcriptional regulatory potential of NACs. Stress-triggered DEGs (with over twofold-change and P-adjust < 0.05) from the drought- or heat-treated samples (see Supplementary Fig. 1a) were collected and clustered by ICA with R package

fastICA[22,77]. To construct sub-GRNs for each cluster, NACs annotated by Rice TF Database [https://ricephylogenomics.ucdavis.edu/tf/genInfo.shtml] were assumed to be "regulators" and all DEGs were set as "targets". Then the "regulator−target" relations were evaluated by GENIE3 algorithm using R package GENIE3 with Z-transformed TPM as input and the Random Forest algorithm as regression tree method[78]. The "regulators" with the highest degree of connectivity in the sub-RGN (hub NACs labeled in Fig. 1b) were chosen for further analysis. Secondly, data on 210 rice cultivars with well-quantified i-traits were used for genetic association analysis on the genomic regions of the hub NACs (see Supplementary Data 1 for detailed information)[20]. Genetic variations among the 210 cultivars were extracted from RiceVarMap2.0[21]. Region-based association analysis was performed by the mixed linear model (MLM) in Tassel 5 software[79] with a threshold of P-value < 1e − 04 to screen the significantly associated variations of a particular trait. Linkage disequilibrium (LD) of the variations within the tested region was calculated by Tassel 5 (expressed as $R^2$). Hub NACs with at least one variation significantly associated to at least one drought-related i-trait were preserved for subsequent analysis. Thirdly, genetic variations and longitude/latitude information of 4720 rice cultivars (including the 210 cultivars) and 446 accessions of common wild rice *Oryza rufipogon* (*O. ruf*) were extracted from RiceVarMap2.0[21] and OryzaGenome[80] for genetic diversity and geographic distribution analysis. Nucleotide diversity (π) in 200-bp windows was extracted by VCFtools (v0.1.16)[81]. The average precipitation data during the years 1979–2015 was extracted from the National Oceanic and Atmospheric Administration (NOAA, [https://www.noaa.gov]). NACs with potential selective sweep (with lower nucleotide diversities in *O. japonica* and *O. indica* compared to *O. ruf*) and i-trait-associated variation(s) that showed precipitation-related geographic distribution were identified as key stress-related NACs.

## Statistics and visualization
If not specified, R (v3.6.1; https://cran.r-project.org/) was used for statistical analysis and plot generation. Two-tailed Student's t test and ANOVA−HSD test was performed using the "t.test" and "aov" coupled with "TukeyHSD" functions (with default parameters), respectively. The high-throughput sequencing tracks were visualized by Integrative Genomics Viewer (IVG; https://igv.org/). The network layouts were generated by Cytoscape (v3.8.2)[82].

## Reporting summary
Further information on research design is available in the Nature Portfolio Reporting Summary linked to this article.

# Data availability
The raw sequencing reads as well as the processed data files generated during this study were deposited into the National Center for Biotechnology Information (NCBI) Gene Expression Omnibus (GEO) under the accession code GSE183241. The IP-MS data have been deposited to the ProteomeXchange Consortium via the PRIDE partner repository with the dataset identifier PXD046729. The public phosphoproteome data of drought-stress-treated HY73 (Hanyou73, *Oryza sativa* ssp. *indica*) can be found in ProteomeXchange under the accession code PXD031505. The minimum dataset required to reproduce the figures in this article have been deposited to the Figshare database[83] [https://doi.org/10.6084/m9.figshare.26039821]. Sequence data of genes mentioned in this article can be found in the Rice Genome Annotation Project or EMBL data libraries under the following accession numbers: *ONAC023* (LOC_Os02g12310) [http://rice.uga.edu/cgi-bin/ORF_infopage.cgi?orf=LOC_Os02g12310.1], *OsREM1.5* (LOC_Os04g45070) [http://rice.uga.edu/cgi-bin/ORF_infopage.cgi?orf=LOC_Os04g45070.1], *OsIMP-α1a* (LOC_Os01g14950) [http://rice.uga.edu/cgi-bin/ORF_infopage.cgi?orf=LOC_Os01g14950.1], *OsIMP-α1b* (LOC_Os05g06350) [http://rice.uga.edu/cgi-bin/ORF_infopage.

cgi?orf=LOC_Os05g06350.1], *SNAC1* (LOC_Os03g60080) [http://rice.uga.edu/cgi-bin/ORF_infopage.cgi?orf=LOC_Os03g60080.1], *OsPYL1* (LOC_Os10g42280) [http://rice.uga.edu/cgi-bin/ORF_infopage.cgi?orf=LOC_Os10g42280.1], *OsFKBP20-1b* (LOC_Os01g62610) [http://rice.uga.edu/cgi-bin/ORF_infopage.cgi?orf=LOC_Os01g62610.1], *OsSF3B1* (LOC_Os02g05410) [http://rice.uga.edu/cgi-bin/ORF_infopage.cgi?orf=LOC_Os02g05410.1], *PGL3* (LOC_Os03g03990) [http://rice.uga.edu/cgi-bin/ORF_infopage.cgi?orf=LOC_Os03g03990.1], *OsPIP2;7* (LOC_Os09g36930) [http://rice.uga.edu/cgi-bin/ORF_infopage.cgi?orf=LOC_Os09g36930.1], *OsANN1* (LOC_Os02g51750) [http://rice.uga.edu/cgi-bin/ORF_infopage.cgi?orf=LOC_Os02g51750.1], *AtREM1.3* (AT2G45820) [https://plants.ensembl.org/Arabidopsis_thaliana/Gene/Summary?g=AT2G45820;r=2:18862953-18864741;t=AT2G45820.1], *SPS* (LOC_Os01g69030) [http://rice.uga.edu/cgi-bin/ORF_infopage.cgi?orf=LOC_Os01g69030.1], *StREM1.3* (DN921712) [https://www.ebi.ac.uk/ena/browser/view/DN921712], *Ubiquitin* (LOC_Os03g13170) [http://rice.uga.edu/cgi-bin/ORF_infopage.cgi?orf=LOC_Os03g13170.1]. Source data are provided with this paper.

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

## Acknowledgements

This study was supported by funds from the National Natural Science Foundation of China (U21A20207, 31930080, 31821005, and 32101666), the Key Research and Development Program of China

(2022YFF1001604), the earmarked fund for the China Agriculture Research System (CARS-01-01) of China, the China Postdoctoral Science Foundation (2019M662657), and the Innovative Project of Hubei Hongshan Laboratory (2022hszd015). We thank the high-throughput phenotyping platform of Huazhong Agricultural University (HZAU) for the analysis of i-traits. The authors acknowledge the HPC platform of the National Key Laboratory of Crop Genetic Improvement for providing the computational resources for all the bioinformatic analysis in this study. Y.C. wishes to thank for the support from Baichuan Fellowship held by the College of Life Science and Technology, HZAU. Y.F. wishes to thank for the support from the Top Talent Support Program of Yangzhou University.

## Author contributions

Y.C. designed and conducted most of the experiments and all bioinformatic analysis, interpreted the data, prepared the figures, and wrote the main parts of the manuscript. Y.F. initiated the project, constructed most of the transgenic materials, participated in part of the experiments and manuscript writing, and revised the manuscript. J.L. and T.Y. participated in the phenotyping experiments. X.L. provided technical support on the immunofluorescence assay. H.T. and Y.Y. provided support on genetic data processing. Y.W. provided technical support in microscopic imaging. L.X. designed and supervised the project and revised the manuscript.

## Competing interests

The authors declare no competing interests.
