## [Peer Review File · Nature Communications]

REVIEWER COMMENTS

Reviewer #1 (Remarks to the Author):

As sessile organisms, plants are frequently subjected to abiotic stresses such as drought and heat stresses, which severely limits plant growth and reduces crop yield. Despite the great efforts made in stress-resilient characterization, the molecular mechanism underlying both drought and heat tolerance remains poorly understood. The manuscript by Chang et al identified ONAC023 as a key regulator of both drought and heat tolerance in rice. The expression of ONAC023 can be induced by stress stimuli, and ONAC023 also exhibits cytoplasm-nucleus transport which depends on dephosphorylated OsREM1.5. Under drought or heat stress, ONAC023 can target and promote the expression of diverse genes functioning in diverse physiological processes. The authors provided lot of experiment evidences to support their findings. However, they could need to address and response the following questions before its possible publication.

1. In Fig 2a-b and Fig 3a, the mutant-complementation plants (onac023 ONAC023) only partially rescued the mutant onac023 phenotypes instead of completely rescued. What is the possible reasons?

2. In Fig 1f, the expression of ONAC023 is induced by salt stress. In addition, ROS accumulation and osmotic stress are also caused by salt stress. Whether ONAC023 and OsREM1.5 also confer resistance to salt stress in rice?

3. ANOVA with Tukey's Honest significant difference (HSD) test is recommended to use to determine P-values in Fig 2a-b.

4. The survival rate of ONAC023-OE osrem1.5 and onac023 osrem1.5 double mutant under drought and heat stress are recommended to investigate, in order to confirm the genetic relationship of ONAC023 and OsREM1.5.

5. In line 179, 'ONAC023 also confers heat tolerance in terms of pollen fertility'. However, the results of RNA-seq and ChIP-seq analyses did not show the DEGs and ONAC023-TGs related to pollen fertility. The authors may want to add this part.

Reviewer #2 (Remarks to the Author):

In this manuscript, Chang and colleagues describe a novel stress-responsive NAC gene associated with drought and heat tolerance in rice at both seedling and reproductive stages. ONAC023 exhibits cytoplasm-nucleus transport via a remorin-importin- α -assisted pathway regulated by the stress-dependent dephosphorylation of OsREM1.5 and the nuclear ONAC023 can target and promote the expression of diverse genes to positively regulate drought and heat tolerance through the integration of multiple stress-responsive processes. These findings provide not only an underlying connection between drought and heat responses, but also a promising candidate gene for multi-stress-resilient rice breeding. I do have some comments that the authors should consider before publication.

Major comments:

Q1 : Although the statistical data of transgenics and mutants can be used to judge the increase of heat and drought resistance of NAC023 to rice, can the number of statistical samples be increased to more support this conclusion?

Q2 : I suggest that the Figure4b adds the total protein checked by western blot. Readers may want to know whether drought or heat influences the total protein level. As shown in Figure 6c, after 45°C 60 min, it is likely that the whole signal was weak compared with 25°C except the GFP control, while the signal in Figure 4a treated with 45°C 3 h was strong. Please explain it.

Q3 : The major conclusion that the activity of ONAC023 in conferring drought and heat stresses depends on OSREM1.5-mediated phosphorylation is not supported by genetic evidence. To test this hypothesis, it's suggested to generate the transgenic materials proUbi-ONAC023-FLAG×proOsREM1.5-OsREM1.5 (-OsREM1.5AAA or -OsREM1.5DDD)/osrem1.5 to detect whether the subcellular location of ONAC023 is directly regulated by OsREM1.5, and evaluated its drought and heat tolerance. This genetic evidence should be informative to elucidate the in vivo effect of OSREM1.5 on ONAC023 protein function.

Q4 : In Fig7, a phos-tag assay can be performed to confirm the phosphorylative state of OsREM1.5 under normal or stress condition including heat.

Q5: Though REM1.5 respond to both drought and heat, it exhibit quite different resistances to these two stresses (fig6b), and only heat was tested in the figure7, so it's hard to judge REM1.5 respond to these two stresses with the same mechanism or a different one.

Q5 : In the final model, the authors seem to propose that the nanodomain is connected to the subcellular translocation of ONAC23, but actually the study did not really provide any direct evidence to couple them.

Minor concerns:

Q1 : In Fig1, the authors classified 98 varieties into the InDel+ and InDel- variety groups for the drought treatment, which supported the evidence that NAC023 enhance the drought resistance of rice. How about the heat resistance of these materials?

Q2 : Was the heat stress treatments performed at night or during the day?

Q3 : There seems to be some differences between the onac023 mutants and wild DJ-WT phenotype before stress in F3b. Did the authors do any statistics for it?

Q4 : The supplementary Fig4 does not provide statistical results of ONAC023-RNAi#2.

Q5 : In the Fig3b experiment, the treated temperature was 43°C, why not stick to the 45°C throughout the manuscript?

Q6 : In Fig3d, the authors may argue that the ONAC023-OE plants at the ripening stage suffered high temperature showed the better resistance than that of ZH11-WT lines in the field, However, this conclusion requires more convincing data, e.g one site over couple of years or one year over couple of sites, ideally couple of years over couple of sites, to rule out other factors.

Q7 : It requires statistics for OsREM1.5 mutant reproductive stage trait in Fig6.

Q8 : The authors claimed that ONAC23 is a hub gene for both drought and heat stresses, so did the authors do any assay to test if rice plants perform better under drought and heat stresses at the same time?

Q9 : Based on fig5c, the BiFC assay showed that REM1.5 interacts with ONAC23 and forms a ring around the nucleus, while in Fig 7f, the mutant version of OsREM1.5 seems to interact with ONAC23, but exhibit a different localization. How do the authors interpret these differences?

Q10 : Whether ONAC023 protein level is directly regulated by phosphorylation, thus influence its localization upon stress treatment?

Q11 : Results, line 257 and Line 282. How to distinguish the plasma membrane and cytosol in tobacco?

Q12 : The authors have done a lot of work, but I think the article can be more concise to make it more readable.

Q13 : OsNTL3 was reported to important for heat stress in rice, any connection between ONAC23 and OsNTL3?

Q14 : The manuscript used quite a lot of the word “significant”, why not just simply specify the foldchange to make it more straightforward.

Response to reviewers:

REVIEWER COMMENTS

Reviewer #1 (Remarks to the Author):

As sessile organisms, plants are frequently subjected to abiotic stresses such as drought and heat stresses, which severely limits plant growth and reduces crop yield. Despite the great efforts made in stress-resilient characterization, the molecular mechanism underlying both drought and heat tolerance remains poorly understood. The manuscript by Chang et al identified ONAC023 as a key regulator of both drought and heat tolerance in rice. The expression of ONAC023 can be induced by stress stimuli, and ONAC023 also exhibits cytoplasm-nucleus transport which depends on dephosphorylated OsREM1.5. Under drought or heat stress, ONAC023 can target and promote the expression of diverse genes functioning in diverse physiological processes. The authors provided lot of experiment evidences to support their findings. However, they could need to address and response the following questions before its possible publication.

Q1: In Fig 2a-b and Fig 3a, the mutant-complementation plants (*onac023 ONAC023*) only partially rescued the mutant *onac023* phenotypes instead of completely rescued. What is the possible reasons?

A1: Thank you for your valuable comments and suggestions. The partial complementation issue of the mutant-complementation plants is not unusual, at least in rice studies. As in the cases of the breeding-valuable gene *NARROW LEAF 1 (NAL1)* (Li et al., 2023) and the anther-dehiscence-decisive gene *FT-INTERACTING PROTEIN 7 (OsFTIP7)* (Song et al., 2018), the traits of interests were partially rescued by the complementation plants but with substantial differences compared to those of the mutants (with $P < 0.05$ by ANOVA or *t*-test). In our article, although the average values of traits for the *onac023 ONAC023* plants are between the values for *onac023* and DJ-WT, but the differences between *onac023 ONAC023* and DJ-WT are not statistically significant (with P -values > 0.05 , see Fig. 2 and Fig. 3) when tested by one-way ANOVA. We speculate that the minor differences between *onac023 ONAC023* and the DJ-WT could be caused by two reasons. On the one hand, the transgenic complementation fragment may not be inserted in a perfectly favourable site in the genome, thus, it may be unexpectedly regulated by nearby *cis*-elements. Also, the epigenetic and/or

three-dimensional genome regulations on the original gene are very likely lost to the complementation fragment. On the other hand, complementation plants suffered additional transgenic process to the mutant and the derived WT plants, which may introduce random mutations to the genome and lead to minor effects on the phenotype. We combined the plants from two independent mutant-complementation lines and used as one in this study to overcome these issues. Therefore, we believe the minor difference between *onac023* *ONAC023* and *onac023* is acceptable and does not affect our conclusion.

Q2: In Fig 1f, the expression of *ONAC023* is induced by salt stress. In addition, ROS accumulation and osmotic stress are also caused by salt stress. Whether *ONAC023* and *OsREM1.5* also confer resistance to salt stress in rice?

A2: We tested the salt stress tolerance of the *ONAC023*-OE by adding 200 mM NaCl in the liquid culture medium. After the 4-day treatment and 7-day recovery, *ONAC023*-OE showed higher survival rate than ZH11-WT (Response Fig. 1). Thus, *ONAC023* may also contribute to salt stress response. Limited by the article space, we do not include additional results on salt stress, and we will get insight into the mechanism of *ONAC023*-mediated salt response in the follow-up research. To better focus on the topic of this study, we removed the qPCR result of salt stress and phytohormone treatments in Fig. 1.

Response Fig. 1: Salt tolerance evaluation on *onac023*.

a Phenotypes of the 14-day old *ONAC023*-OE and ZH11-WT seedlings under the salt stress treatment in the liquid culture medium. **b** Survival rate statistics of the seedlings after the recovery. Cross bars indicate mean \pm SEM. *P*-values were determined by one-way ANOVA–HSD test ($P < 0.05$).

Q3: ANOVA with Tukey's Honest significant difference (HSD) test is recommended to use to determine *P*-values in Fig 2a-b.

A3: Thank you for your suggestion. We have replaced the statistical method with ANOVA–HSD test for all the multi-group comparisons in the article (including those in Fig. 2).

Q4: The survival rate of *ONAC023*-OE *osrem1.5* and *onac023 osrem1.5* double mutant under drought and heat stress are recommended to investigate, in order to confirm the genetic relationship of *ONAC023* and *OsREMI.5*.

A4: We started the construction of the recommended transgenic materials once we received the comments, however, it requires at least nine months to get the T₁ generation plants with stable expression for further experiments. As a workaround, we generated the *ONAC023*-OE *osrem1.5* and *onac023 osrem1.5* cell lines by transient transformation of the *osrem1.5* protoplasts and evaluated their ROS accumulation levels in response to osmotic (by 30% PEG 4000 treatment for 25 min) or heat (45°C for 30 min) stress by H₂DCFDA fluorescence staining. The result shows that *onac023* and *onac023 osrem1.5* cells accumulated matching levels of ROS which were significantly higher than that of the *ONAC023*-OE cells, while overexpression of *ONAC023* in the *osrem1.5* cells can hardly (less than 1.5 foldchanges of the H₂DCFDA fluorescence intensity) rescue the ROS level (Fig. 7e, f). In addition, the co-expression of *ONAC023* and *OsREMI.5^{AAA}*, but not *ONAC023* and *OsREMI.5^{DDD}*, in the *osrem1.5* cells led to decreased ROS accumulation compared with the situation in *osrem1.5* and *osrem1.5 onac023* cells (Fig. 7e, f). Since ROS scavenging is one of the mechanisms of *ONAC023*-integrated stress regulation, we believe these results can, at least in part, confirm the genetic relationship of *ONAC023* and *OsREMI.5*.

Q5: In line 179, 'ONAC023 also confers heat tolerance in terms of pollen fertility'. However, the results of RNA-seq and ChIP-seq analyses did not show the DEGs and *ONAC023*-TGs related to pollen fertility. The authors may want to add this part.

A5. We did show the evaluation of a heat stress DEG *OsAPT2* (LOC_Os04g42520) that determines the thermo-sensitive male sterility phenotype in 'Annong S-1' (the first discovered thermo-sensitive genic male sterility mutant in rice) (Zhou et al., 2006) in

Supplementary Fig. 20b. Binding of ONAC023 to the ~+200 bp region of *OsAPT2* TSS was also identified by the ChIP-seq (Supplementary Fig. 20e). In the revision, we further validated the differential expression and ONAC023-targeting of *OsAPT2* in the flowering inflorescence under heat stress (43°C). The qPCR results show that mutation of *ONAC023* also decreases *OsAPT2* expression level in the panicle, and the binding of ONAC023 can still be detected by ChIP-qPCR at the peak region of the seedling ChIP-seq (Supplementary Fig. 20e). Although the binding site is slightly out of the -3000 – +100 bp range for a promoter, our results indicated that *OsAPT2* is at least regulated by ONAC023 in the inflorescence under heat stress. Since the decreased expression of *OsAPT2* led to male sterility (Zhou et al., 2006), this result can partially explain the essentiality of ONAC023 to pollen fertility under heat stress.

Reviewer #2 (Remarks to the Author):

In this manuscript, Chang and colleagues describe a novel stress-responsive NAC gene associated with drought and heat tolerance in rice at both seedling and reproductive stages. ONAC023 exhibits cytoplasm-nucleus transport via a remorin-importin- α -assisted pathway regulated by the stress-dependent dephosphorylation of OsREM1.5 and the nuclear ONAC023 can target and promote the expression of diverse genes to positively regulate drought and heat tolerance through the integration of multiple stress-responsive processes. These findings provide not only an underlying connection between drought and heat responses, but also a promising candidate gene for multi-stress-resilient rice breeding. I do have some comments that the authors should consider before publication.

Major comments:

Q1: Although the statistical data of transgenics and mutants can be used to judge the increase of heat and drought resistance of NAC023 to rice, can the number of statistical samples be increased to more support this conclusion?

A1: We greatly appreciate your professional review work on our manuscript and valuable comments and suggestions. The data points shown in the statistical analysis do not denote individual plants, but pots each with at least 15 seedlings or randomized blocks each with at least 10 plants, which we believe is an acceptable sample size for abiotic stress-focusing

studies in rice. The main limit for using larger number of blocks or larger field is the concerns for environmental conditions control. For drought stress, the soil moisture can vary greatly among blocks, even though the water supply is cut off simultaneously for all the blocks in the field. Thus, only the blocks with similar soil moisture during the drought treatment process were included for traits evaluation. For heat stress under natural field conditions, the exact occurrence of peak temperature is highly unpredictable, thus, we performed interval sowing in the blocks and only collected the data from the blocks whose flowering date overlapping with the period of peak temperature. In this regard, we prefer that traits evaluation on the blocks with careful control of the environmental conditions is more persuasive than the bulk data from plants in a large but less controllable field. To further support our conclusion, we have added the result of drought tolerance evaluation on the *ONAC023*-OE lines in Hainan province, PRC (18°32'34" N, 110°2'7" E) in the revision. Therefore, both in-field drought tolerance and heat tolerance evaluations are now supported with multi-year/-location data.

Q2: I suggest that the Figure 4b adds the total protein checked by western blot. Readers may want to know whether drought or heat influences the total protein level. As shown in Figure 6c, after 45°C 60 min, it is likely that the whole signal was weak compared with 25°C except the GFP control, while the signal in Figure 4a treated with 45°C 3 h was strong. Please explain it.

A2: We have included the result of western blot with total protein extracts in Fig. 4b. We did not observe noticeable trace of protein accumulation or degradation under stress treatments. The fluorescent signal seems to be weaker in the *N. benthamiana* cells after 45°C for 60 min, which might be due to the slight differences among the cell states or gene expression levels for different transfections. That is the reason why we avoided the comparisons of absolute fluorescence intensities across different transfections. Instead, we did all the fluorescence intensity-related analysis (mostly on subcellular localization but not protein quantification) only after the normalization of fluorescent signals to the maximum value in the sampling region (see Fig. 4a and Fig. 5e). It is also possible that the cells we chose to exhibit in the figure were overstressed and the fluorescent signal can be decreased for a dying cell. In fact, the 45°C treatment for over 60 min often resulted in the failure of detecting clear fluorescent

signals, even though the fusion protein can still be detected by Western blot (see Response Fig. 2). To avoid misunderstanding of our result, we replaced the images with another replicate showing matching fluorescent intensities (in the nuclei) to the 45°C 30 min and 25°C slices in the revised Fig. 6. We believe both replicates serve the purpose of detecting the subcellular localization changes of ONAC023 and OsREM1.5 under stress.

Response Fig. 2: Decreased fluorescent signal but not protein level under extreme stress in *N. benthamiana* cells.

a Slices of the ONAC023-FLAG-GFP expressing *N. benthamiana* cells under 25°C and 45°C for 75 min. Scale bar = 20 μ m. **b** Western blot bands of the ONAC023-FLAG-GFP proteins extracted from the 1 cm \times 1 cm slices of transfected leaves in **a**. The bands were detected by the FLAG-tag fused to the N terminal of ONAC023-GFP.

Q3: The major conclusion that the activity of ONAC023 in conferring drought and heat stresses depends on OSREM1.5-mediated phosphorylation is not supported by genetic evidence. To test this hypothesis, it's suggested to generate the transgenic materials *proUbi-ONAC023-FLAG* \times *proOsREM1.5-OsREM1.5* (*-OsREM1.5^{AAA}* or *-OsREM1.5^{DDD}*)/*osrem1.5* to detect whether the subcellular location of ONAC023 is directly regulated by OsREM1.5, and evaluated its drought and heat tolerance. This genetic evidence should be informative to elucidate the in vivo effect of OSREM1.5 on ONAC023 protein function.

A3: Since the generation of rice transgenic lines with stable expression (at least the T₁ generation) requires at least nine months and cannot be accomplished within the revision period (3 months) even some extension. Therefore, we chose an alternative approach by transfecting the *osrem1.5* protoplasts with *ProUbi::ONAC023-mcherry* × *ProOsREM1.5::OsREM1.5^{AAA}/OsREM1.5^{DDD}*. The result shows that the introduction of *OsREM1.5^{AAA}*, but not *OsREM1.5^{DDD}*, can restore the nuclear localization of ONAC023–mcherry under the osmotic stress (by 30% PEG 4000 treatment for 25 min) and heat stress (45°C for 30 min) (Fig. 6e, f). We further evaluated the ROS accumulation levels of these cell lines in response to the osmotic stress and heat stress by H₂DCFDA fluorescence staining. The result shows that the co-expression of *ONAC023* and *OsREM1.5^{AAA}*, but not *OsREM1.5^{DDD}*, in the *osrem1.5* cells led to decreased ROS accumulation compared with the *osrem1.5* and ZH11-WT. Since ROS scavenging is one of the mechanisms of ONAC023-integrated stress regulation, we believe these results can, at least in part, confirm the genetic relationship of *ONAC023* and *OsREM1.5*.

Q4: In Fig 7, a phos-tag assay can be performed to confirm the phosphorylative state of OsREM1.5 under normal or stress condition including heat.

A4: We performed a phos-tag assay with the immunoprecipitated OsREM1.5–HA from the transfected *N. benthamiana* epidermal cells. The result shows that the bands for phosphorylated OsREM1.5 can be detected under the normal condition, but not under the drought or heat stress conditions (Fig. 7b). Mutations of S80T92S97 to A80A92A97 (OsREM1.5^{AAA}) aborted the phosphorylation bands under the normal condition. Thus, OsREM1.5 is phosphorylated at the S80T92S97 site under normal but not under drought or heat stress conditions, and the phosphorylation on this site is the primary form of phosphorylated OsREM1.5. We also noticed that only a fraction of OsREM1.5 was phosphorylated under the normal condition. This might be due to the dynamic nature of protein phosphorylation state.

Q5: Though *REM1.5* respond to both drought and heat, it exhibit quite different resistances to these two stresses (fig 6b), and only heat was tested in the figure 7, so it's hard to judge

REM1.5 respond to these two stresses with the same mechanism or a different one.

A5: We have added the slices for drought/osmotic treatment (15% PEG 6000 treatment in the water media) in Fig. 6 and 7 in the revision. The slices exhibited similar subcellular localizations of *OsREM1.5* and *ONAC023* to those under the heat stress. It is possible that *OsREM1.5* responses to the two conditions with similar mechanism.

Q6: In the final model, the authors seem to propose that the nanodomain is connected to the subcellular translocation of *ONAC023*, but actually the study did not really provide any direct evidence to couple them.

A6: The initial model was proposed based on the results that the nuclear translocation of *ONAC023* co-occurs with the plaque-like co-localization of *ONAC023* and *OsREM1.5*, and the formation of plaque-like spots, as well as the translocation of *ONAC023*, requiring the complete dephosphorylation of *OsREM1.5*. Thus, the plaque-like spots, potentially *OsREM1.5*-integrated nanodomains, might be the structure basis for the translocation of *ONAC023*. However, we agree that readers may find these relationships not direct enough especially when we cannot provide the detailed composition of the plaque-like spots or the trace of *ONAC023* at single-molecule level, due to the limitation of current techniques. Therefore, we removed the conclusions on the nanodomain in the final model and made discussion on its possibility as a structural basis for *ONAC023* translocation in the Discussion section in the revised manuscript.

Minor concerns:

Q1: In Fig 1, the authors classified 98 varieties into the InDel^+ and InDel^- variety groups for the drought treatment, which supported the evidence that *NAC023* enhance the drought resistance of rice. How about the heat resistance of these materials?

A1: In another ongoing project, we have evaluated the seed-setting rates of another rice population under heat stress and the data is available for 55 out of the 98 varieties. As exhibited in Response Fig. 3, the InDel^+ varieties also obtained higher seed-setting rates compared with the InDel^- under the heat stress. Since the data is unpublished and its absence does not influence the conclusion of this study, the result is currently not included in this article.

Response Fig. 3: The seed setting rates of the 55 out of the 98 tested varieties in Fig. 1 after the heat stress treatment at the flowering stage (with the air temperature of 43°C for 3 h per day at noon during flowering).

The *P*-value is determined by two-tailed Student's *t*-test.

Q2: Was the heat stress treatments performed at night or during the day?

A2: For all the seedling samples, the heat stress treatments began at 8:00 am in a growth chamber with constant light (15,000 lux) during the whole treatment process. For heat stress treatments at the flowering stage, heat stress took effect during the noon time (11:00 am – 3:00 pm). We have clarified these in the Methods section.

Q3: There seems to be some differences between the *onac023* mutants and wild DJ-WT phenotype before stress in F3b. Did the authors do any statistics for it?

A3: We have added the phenotype photos and trait evaluation result of *onac023* and DJ-WT under the normal condition in Supplemental Fig. 5. The result indicates that the differences in yield, seed-setting, and plant height between *onac023* and DJ-WT are minor ($P > 0.05$ by two-tailed Student's *t*-test) under normal conditions (Supplemental Fig 7a). For in-field drought treatment, we define the relative moisture under 40% at the 20 cm depth under the ground as the starting point of drought stress (when the “before drought” photo was taken). In fact, this criterion might be a mild stress condition for drought hypersensitive plants like

onac023 (water in the ground surface was completely evaporated). This might be part of the reasons for the differences between the *onac023* mutants and wild DJ-WT before stress shown in the picture.

Q4: The supplementary Fig 4 does not provide statistical results of *ONAC023*-RNAi#2.

A4: We are sorry for the negligence. We have included the statistical results of *ONAC023*-RNAi#2 in the revised Supplemental Fig 4.

Q5: In the Fig 3b experiment, the treated temperature was 43°C, why not stick to the 45°C throughout the manuscript?

A5: We used 43°C for panicle-involved tests because the rice inflorescence is more vulnerable to heat stress compared with the 4-leaf stage seedlings. For japonica rice like ZH11 and DJ, heat treatment with 45°C often leads to less significant results or complete sterility for both transgenic lines and WT. We have made brief explanation on this issue in the revised Method and Material section.

Q6: In Fig 3d, the authors may argue that the *ONAC023*-OE plants at the ripening stage suffered high temperature showed the better resistance than that of ZH11-WT lines in the field, However, this conclusion requires more convincing data, e.g one site over couple of years or one year over couple of sites, ideally couple of years over couple of sites, to rule out other factors.

A6: Fig. 3d includes the data of the 2-year test in Wuhan, and we labelled the exact years of the tests in the revised figure so that readers won't miss this information. We only perform the multi-year test in the Wuhan field because the peak temperature in other available locations is neither suitable nor predictable due to the lack of precise meteorological data. In addition, the 2-year test was performed in two different fields with > 1 km distance to exclude any other different factors such as soil nutrition conditions that may affect the fertility and yield.

Q7: It requires statistics for *OsREM1.5* mutant reproductive stage trait in Fig 6.

A7: The seed setting rates of the *osrem1.5* lines and the ZH11-WT under normal and stress conditions were added in the revised Fig. 6. Compared with the WT, the mutant lines exhibited no significant difference ($P > 0.05$ by ANOVA test) in seed setting rates under the normal condition but decreased seed setting rates under the drought and heat stress conditions. The result further supports the essential role of *OsREM1.5* in drought and heat stress responses.

Q8: The authors claimed that *ONAC23* is a hub gene for both drought and heat stresses, so did the authors do any assay to test if rice plants perform better under drought and heat stresses at the same time?

A8: We did try the couple stress treatment on the *ONAC023*-OE plants, but simple combination of the drought and heat condition used in this article does not lead to persuasive data, as the average survival rates of both OE and WT plants were close to zero, which supports the conclusion that multifactorial stress combination can lead to very severe damage to plants even if the component stress is mild (as recently reviewed by Zandalinas et al. 2024 in the Plant J). We are still working on finding optimal conditions for couple stress treatment.

Q9: Based on fig 5c, the BiFC assay showed that REM1.5 interacts with ONAC23 and forms a ring around the nucleus, while in Fig 7f, the mutant version of OsREM1.5 seems to interact with ONAC23, but exhibit a different localization. How do the authors interpret these differences?

A9: This difference might be the result of two aspects. On the one hand, rice protoplasts are smaller compared to the tobacco cells so that it cannot provide enough resolution for distinguishing specified localization patterns especially when the protein is overexpressed. This is also the reason why we chose to observe the plaque-like spots in the tobacco cell instead of rice protoplasts. On the other hand, the two parts of the VENUS protein can be sticky once ONAC023 and the dephosphorylated OsREM1.5 interact with each other, thus the interaction is likely to be kept even when OsREM1.5 is dynamically phosphorylated, which can result in continuous localization signal similar to OsREM1.5^{DDD}. Despite the

difference in localization, both BiFC slices support the ONAC023–OsREM1.5/OsREM1.5^{AAA} interactions.

Q10: Whether ONAC023 protein level is directly regulated by phosphorylation, thus influence its localization upon stress treatment?

A10: Since we only discussed the phosphorylation of OsREM1.5, but not ONAC023, in this study, we assume that ONAC023 you mentioned here actually refers to OsREM1.5. We performed an *in vitro* cell free protein degradation assay on the OsREM1.5^{AAA}–His and OsREM1.5^{DDD}–His proteins. The result shows that both forms have similar degradation rates (complete degradation at 30 min of incubation, Response Figure 4). Thus, the phosphorylation on S80T92S97 of OsREM1.5 is likely to have minor (if any) effect on the protein.

Response Fig. 4: OsREM1.5^{AAA}–His and OsREM1.5^{DDD}–His showed comparable degradation rate in the *in vitro* cell free degradation assay.

The assay was performed according to Li et al (2023) with minor modifications (Li et al., 2023). Total protein was extracted from drought and heat stress-treated ZH11 14-d seedling. 50 ng of the purified OsREM1.5^{AAA}–His or OsREM1.5^{DDD}–His was incubated with 500 μ g of the total protein in 250 μ L of degradation buffer (25 mM Tris–HCl, pH 7.5, 10 mM NaCl, 10 mM MgCl₂, 10 mM ATP) at 37°C. Equal volumes of the mixture was collected at the indicated time and detected by Anti-His western blot. Repeat of the assay with 500 μ g of denatured (100°C for 15 min) total protein was used as control.

Q11: Results, line 257 and Line 282. How to distinguish the plasma membrane and cytosol in tobacco?

A11: Though the two signals are close to each other in tobacco cells, the fluorescent signal from the plasma membrane is generally smoother and more concentrated (or thinner) than that from the cytosol. More importantly, even if this difference is hard to be distinguished by visual observation for some parts of the cell, the peak intensities of the two signals are not converged as indicated by the dashed lines in the fluorescence profile plot of Fig. 6d. We have added brief explanations to help readers judge the differences in localizations by the profile plot.

Q12: The authors have done a lot of work, but I think the article can be more concise to make it more readable.

A12: In the revised manuscript, we reorganized our story in a more “straightforward” way. Unnecessary explanations of results that can be easily read from figures have been removed. And the writing has been revised for the Discussion section. This work involves multifaced approaches to reveal the unique and multi-layered regulation of ONAC023, and some part of the results may be comprehensive. We have tried our best to present the data more readable, and hope the revised version will be acceptable.

Q13: OsNTL3 was reported to important for heat stress in rice, any connection between ONAC23 and OsNTL3?

A13: Both ONAC023 and OsNTL3 are stress-responsive NACs that can translocate into the nucleus and contribute to heat stress tolerance by activating downstream gene expression. However, OsNTL3 is known as a representative of the plasma membrane-anchored NACs that translocate into the nucleus by the cleavage of their transmembrane domains (Liu et al., 2020), while the translocation mechanism is totally different for ONAC023, which does not contain any predictable transmembrane domains and requires the association of OsREM1.5 for translocation. In addition, we compared the ONAC023-targeting genes under the heat stress and the genes up-regulated by OsNTL3. The result shows only 16 overlapped genes between the two gene sets (Response Fig. 5a). Knowledge on the 16 genes is very limited and

most of them are not typical stress-responsive genes according to the predicted functions (Response Fig. 5b). Thus, we think that the downstream pathways regulated by ONAC023 and OsNTL3 are largely different. We have mentioned this point in the Discussion section.

Response Fig. 5: Intersections of the ONAC023-targeting genes and OsNTL3 up-regulated genes under heat stress.

a Venn Diagram showing the intersection genes between ONAC023-targeting genes and genes up-regulated by OsNTL3. **b** Accession IDs and predicted functions of the 16 intersection genes.

Q14: The manuscript used quite a lot of the word “significant”, why not just simply specify the foldchange to make it more straightforward.

A14: We have added the values of fold-changes and/or *P*-values wherever significant differences should be stated in the manuscript as you suggested.

References for review

- Li, W. J., *et al.* Serine protease NAL1 exerts pleiotropic functions through degradation of TOPLESS-related corepressor in rice. *Nat. Plants* **9**, 1130-1142 (2023).
- Liu, X. H., Lyu, Y. S., Yang, W., Yang, Z. T., Lu, S. J., Liu, J. X. A membrane-associated NAC transcription factor OsNTL3 is involved in thermotolerance in rice. *Plant Biotechnol. J.* **18**, 1317-1329 (2020).
- Song, S. Y., Chen, Y., Liu, L., See, Y. H. B., Mao, C. Z., Gan, Y. B., Yu H. OsFTIP7 determines auxin-mediated anther dehiscence in rice. *Nat. Plants* **4**, 495-504 (2018).

Zhou, C., Li, J., Zou, J. C. Liang, F. S., Ye, C. J., Jin, D. M., Weng, M. L., Wang, B.. Cloning and characterization of a second form of the rice adenine phosphoribosyl transferase gene (*OsAPT2*) and its association with TGMS. *Plant Mol. Biol.* **60**, 365-376 (2006).

Zandalinas S. I., Cassal, J., Rouached, H, Mittler, R. Stress combination: from genes to ecosystems. *Plant J.* **117**, 1639-1641 (2024)

REVIEWERS' COMMENTS

Reviewer #1 (Remarks to the Author):

The revised manuscript is now substantially improved. Through new experimental data or additional explanation, the authors have responded to all my comments and successfully addressed my concerns. I have no further comments to add.

Reviewer #2 (Remarks to the Author):

The author basically addressed my concerns, no more questions.